# Long non-coding RNA *RAMS11* promotes metastatic colorectal cancer progression

Jessica M. Silva-Fisher [1,2], Ha X. Dang[1,2,3], Nicole M. White[1,2], Matthew S. Strand[4], Bradley A. Krasnick[4], Emily B. Rozycki[1], Gejae G. L. Jeffers[1], Julie G. Grossman [4], Maureen K. Highkin[1], Cynthia Tang [1], Christopher R. Cabanski[5], Abdallah Eteleeb[1], Jacqueline Mudd[4], S. Peter Goedegebuure[4], Jingqin Luo[2,6], Elaine R. Mardis [7], Richard K. Wilson[7], Timothy J. Ley [1,2], Albert C. Lockhart[8], Ryan C. Fields[2,4,10] & Christopher A. Maher [1,2,3,9,10 ✉]

Colorectal cancer (CRC) is the most common gastrointestinal malignancy in the U.S.A. and approximately 50% of patients develop metastatic disease (mCRC). Despite our understanding of long non-coding RNAs (lncRNAs) in primary colon cancer, their role in mCRC and treatment resistance remains poorly characterized. Therefore, through transcriptome sequencing of normal, primary, and distant mCRC tissues we find 148 differentially expressed RNAs Associated with Metastasis (*RAMS*). We prioritize *RAMS11* due to its association with poor disease-free survival and promotion of aggressive phenotypes in vitro and in vivo. A FDA-approved drug high-throughput viability assay shows that elevated *RAMS11* expression increases resistance to topoisomerase inhibitors. Subsequent experiments demonstrate *RAMS11*-dependent recruitment of Chromobox protein 4 (CBX4) transcriptionally activates Topoisomerase II alpha (TOP2α). Overall, recent clinical trials using topoisomerase inhibitors coupled with our findings of *RAMS11*-dependent regulation of TOP2α supports the potential use of *RAMS11* as a biomarker and therapeutic target for mCRC.

---

[1] Department of Internal Medicine, Washington University School of Medicine, St. Louis, MO, USA. [2] Alvin J. Siteman Cancer Center, Washington University School of Medicine, St. Louis, MO, USA. [3] The McDonnell Genome Institute, St. Louis, MO, USA. [4] Department of Surgery, Washington University School of Medicine, St. Louis, MO, USA. [5] Parker Institute for Cancer Immunotherapy, San Francisco, CA, USA. [6] Division of Public Health Sciences, Department of Surgery, Washington University School of Medicine, St. Louis, MO, USA. [7] Institute for Genomic Medicine, Nationwide Children's Hospital, Columbus, OH, USA. [8] Department of Medicine, University of Miami, Miami, FL, USA. [9] Department of Biomedical Engineering, Washington University School of Medicine, St. Louis, MO, USA. [10] These authors contributed equally: Ryan C. Fields, Christopher A. Maher. ✉email: christophermaher@wustl.edu

Colorectal cancer (CRC) is the most common gastro-intestinal malignancy in the United States. At the time of initial diagnosis, 20% of patients present with metastasis, and of those patients with primary disease approximately 50% will eventually develop metastatic disease[1]. Furthermore, the overall 5-year survival rate for patients with metastatic CRC (mCRC) is only 14%[2,3]. Currently, there are numerous therapeutic treatments for patients with mCRC including surgery, cytotoxic chemotherapy, targeted therapy, immunotherapy, radiation, and combination strategies. However, there is little pre-treatment data that can predict response to treatment and development of resistance[4]. While there are promising developments in second-line treatment options for mCRC patients using cytotoxic agents or targeted agents, the mechanisms driving metastatic progression remain poorly characterized thus prohibiting effective drug development. Furthermore, response to second-line treatment is even less effective than first line[5,6]. These statistics and poor treatment options highlight the critical need for improved biomarker-driven therapies at the time of diagnosis.

To date, CRC research has primarily focused on the deregulation of protein-coding genes to identify oncogenes and tumor suppressors as potential diagnostic and therapeutic targets[7,8]. While more recent studies have explored the role of microRNAs in CRC[9,10], there is still a lack of studies focusing on long non-coding RNAs (lncRNAs) in mCRC. LncRNAs are typically greater than 200 nucleotides in length, lack coding potential, are transcribed by RNA polymerase II, spliced, 5′ capped, and polyadenylated[11]. LncRNAs are known to have a diverse range of biological functions, including serving as critical regulators in tumorigenesis and metastasis[12–18]. Furthermore, the clinical significance of lncRNAs can be exemplified by their use as diagnostic, prognostic, predictive biomarkers, and potential therapeutic targets[19–24]. Therefore, the characterization of lncRNAs, elucidating their function, and assessing their clinical applicability could significantly impact mCRC diagnosis and treatment.

While transcriptome sequencing has provided an unbiased method for discovering lncRNAs, existing large-scale sequencing projects such as The Cancer Genome Atlas Network (TCGA)[25] are comprised of predominantly primary tumors lacking matched metastatic samples. This represents a critical barrier to discovering novel lncRNAs throughout the progression of primary to metastatic disease correlated to treatment response and resistance. To address this, we have conducted a meta-analysis of normal, primary, and distant metastatic tissues from CRC patients across two independent patient cohorts to discover differentially expressed (DE) lncRNAs in metastatic tumors compared with primary tumors, termed RNAs Associated with Metastasis (RAMS). We have prioritized RAMS11 as it was a top up-regulated lncRNA in metastasis and associated with poor disease-free survival across multiple cohorts. We then demonstrate that RAMS11 promotes aggressive phenotypes in vitro and in vivo. While lncRNAs have been shown to promote tumor progression[26–28], the understanding of their role in treatment resistance is still unknown. Therefore, we have utilized a drug screen to discover that RAMS11 promotes resistance to topoisomerase inhibitors and provide mechanistic insight into RAMS11-dependent topoisomerase II alpha (TOP2α) regulation to promote mCRC.

## Results

### LncRNA landscape of mCRC.
To identify consistently altered lncRNAs during mCRC, we performed transcriptome sequencing and analysis of 37 patients from two independent cohorts. The first cohort includes ten normal colon epithelium, two primary CRC, and fourteen distant mCRC patient samples collected from Washington University, termed WUSTL cohort. The second cohort is from a previously published transcriptome sequencing study by Kim et al.[29] using matched normal, primary, and metastatic samples from 18 CRC patients, termed Kim cohort (Fig. 1a).

To identify lncRNAs altered in the metastatic samples relative to primary and normal samples, we performed a meta-analysis of the WUSTL and Kim cohorts. We identified 148 DE lncRNAs (FDR < 0.05, fold change > 2) in metastasis, termed RAMS (Fig. 1b and Supplementary Data 1). Several previously well-known and characterized lncRNAs known to promote oncogenic phenotypes in CRC or other cancer types were also detected. This includes increased expression of H19, HULC, CCAT4, and TCONS_I2_00022545 in mCRC and decreased expression of FENDRR in metastatic samples[30–34] (Fig. 1b). Overall, this serves as a key meta-analysis from aggressive CRC patient tissues to establish the mCRC lncRNA landscape.

### RAMS11 is upregulated in mCRC.
We prioritized our functional studies on lncRNAs that were highly deregulated and potentially clinically relevant in mCRC. To prioritize all RAMS, we evaluated whether their expression correlated with patient outcome. First, we found that six of the 148 RAMS were associated with disease-free survival using 232 patients from the TCGA CRC cohort (RNA-Seq). Among the six RAMS associated with survival in the TCGA cohort, only RAMS11 was associated with poor survival from a second cohort of 82 patients (Fig. 1a, c) from the Sveen study (GSE24549, exon array[35]). These results indicate that high levels of RAMS11 in primary tumors may serve as an indication of poor patient outcome. Notably, RAMS11 was also a top upregulated lncRNA in metastatic tumors (FPKM = 4.81) as compared with primary tumors (combined $p = 2.56 \times 10^{-10}$ average fold change = 6.1) and normal tissues (combined $p = 2.2 \times 10^{-20}$, average fold change = 12.9) (Fig. 1d and Supplementary Fig. 1a). We further validated the upregulation of RAMS11 by qPCR when comparing matched metastatic patient samples with normal ($p = 0.007$, two-tailed paired $t$-test) and primary ($p = 0.024$, two-tailed paired $t$-test) patient samples (Supplementary Fig. 1b).

Our de novo transcript assembly using the WUSTL cohort identified RAMS11 as a five-exon transcript of 959 nucleotides, which we confirmed by 5′ and 3′ rapid amplification of cDNA ends (RACE) (Fig. 1d, Supplementary Data 2). Previously, three exons of the RAMS11 transcript were annotated as LINC01564 (NR_125841) through a microarray probe-based analysis[36] (Fig. 1d). We further characterized RAMS11 expression in a panel of CRC cell lines. RAMS11 was highly expressed in a panel of six primary (more than three-fold increase) and two mCRC cell lines (more than 11-fold increase) compared with CCD18-Co, a normal colon control cell line (Supplementary Fig. 1c). Since the cellular localization of lncRNAs can help decipher their functions, we fractionated LoVo mCRC cells with high endogenous expression of RAMS11. As shown in Supplementary Fig. 1d, RAMS11 is predominantly expressed in the nucleus (89.5%), with only a 10.5% expression in the cytoplasm. Taken together, these results show that RAMS11 is a five-exon, nuclear localized, lncRNA that is highly expressed in primary and mCRC cell lines and absent in normal colon epithelium.

### RAMS11 promotes aggressive phenotypes in vitro.
To understand RAMS11 functional significance, we created a RAMS11 knockout (KO) model by generating two CRISPR/Cas9 luciferase-tagged cell lines with a genomic deletion of the last four exons of RAMS11 in the LoVo metastatic colon cancer cell line

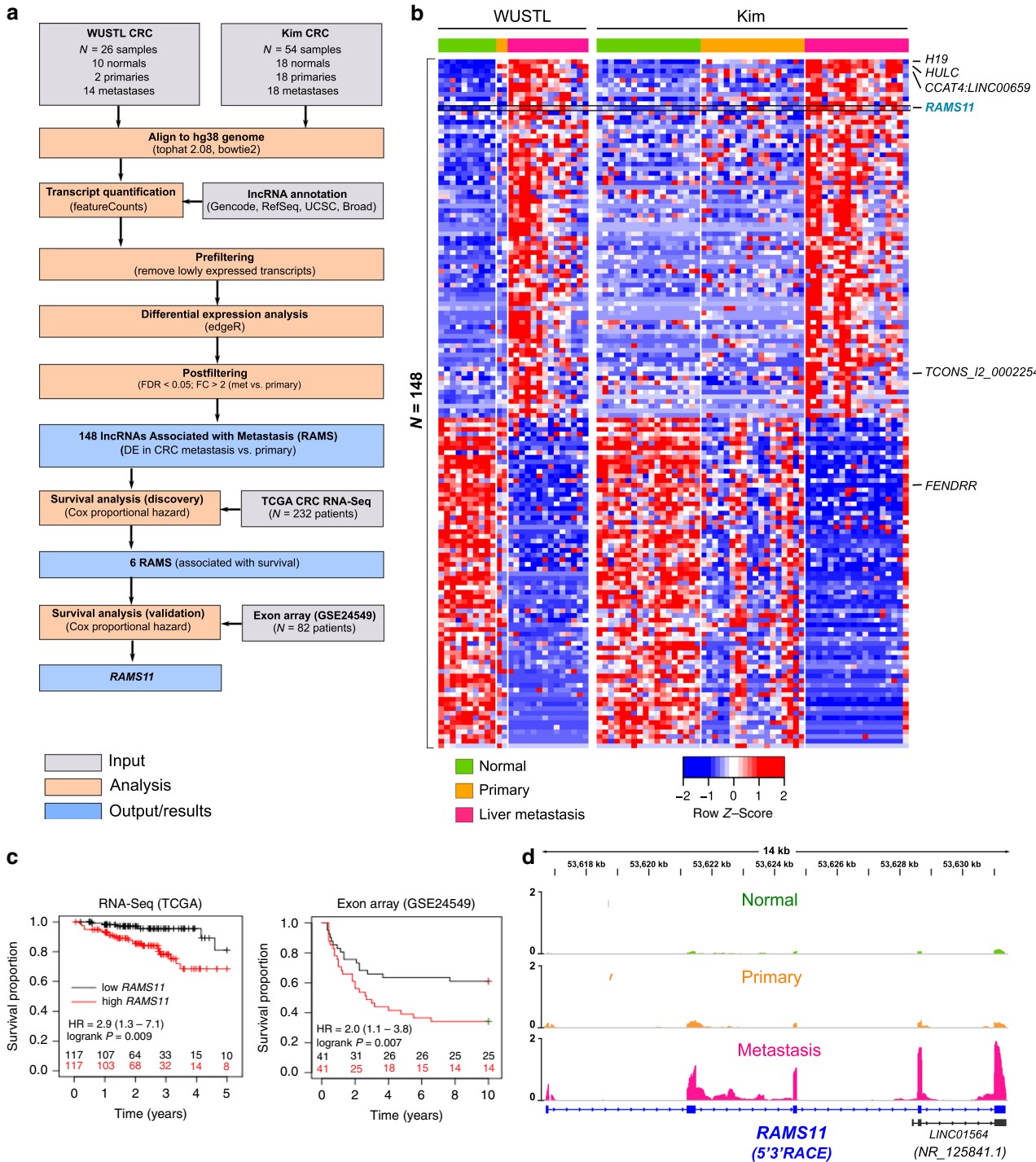

**Fig. 1 RNAs associated with metastasis (RAMS). a** Analysis pipeline for discovery of metastatic CRC lncRNAs. Shaded gray color boxes (input), orange color boxes (analysis), and blue color boxes (output/results). **b** Heatmap of lncRNAs differentially expressed in metastasis compared with primary. Patient samples are indicated on top row shown as normal (green), primary (orange), and liver metastasis (pink). Heatmap color is scaled by row expression Z-score. **c** Kaplan–Meir plots showing RAMS11 association with poor disease-free survival in The Cancer Genome Atlas (TCGA) RNA-Seq and exon array (GSE24549) datasets. Numbers above x-axis are patients at risk at the intervals. p values are inferred from a two-sided logrank test. **d** Average normalized RNA-Seq coverage across WUSTL and Kim cohorts. Normal samples are green boxes, primary samples are orange boxes, and metastatic samples are pink boxes. 5'3' RACE validated five-exon sequence is shown below in blue.

(Supplementary Fig. 2a). We confirmed greater than a 99.9% reduction in our RAMS11 CRISPR KO models (clones referred to as CRISPR1 and CRISPR2) relative to wild-type cells (Fig. 2a) and confirmed that the genomic deletion of RAMS11 did not alter the expression of adjacent genes GCLC and KLH31 (Supplementary Fig. 2b, c).

We used these genetically engineered cell lines to determine changes in the invasiveness of cells using Matrigel-coated transwells in a modified Boyden chamber assay. There was more than a 60% decrease in invasion of RAMS11 CRISPR KO cells (CRISPR1 $p = 0.004$, CRISPR2 $p = 0.023$, two-tailed paired $t$-test) compared with wild-type cells (Fig. 2c, e). We also conducted a transient

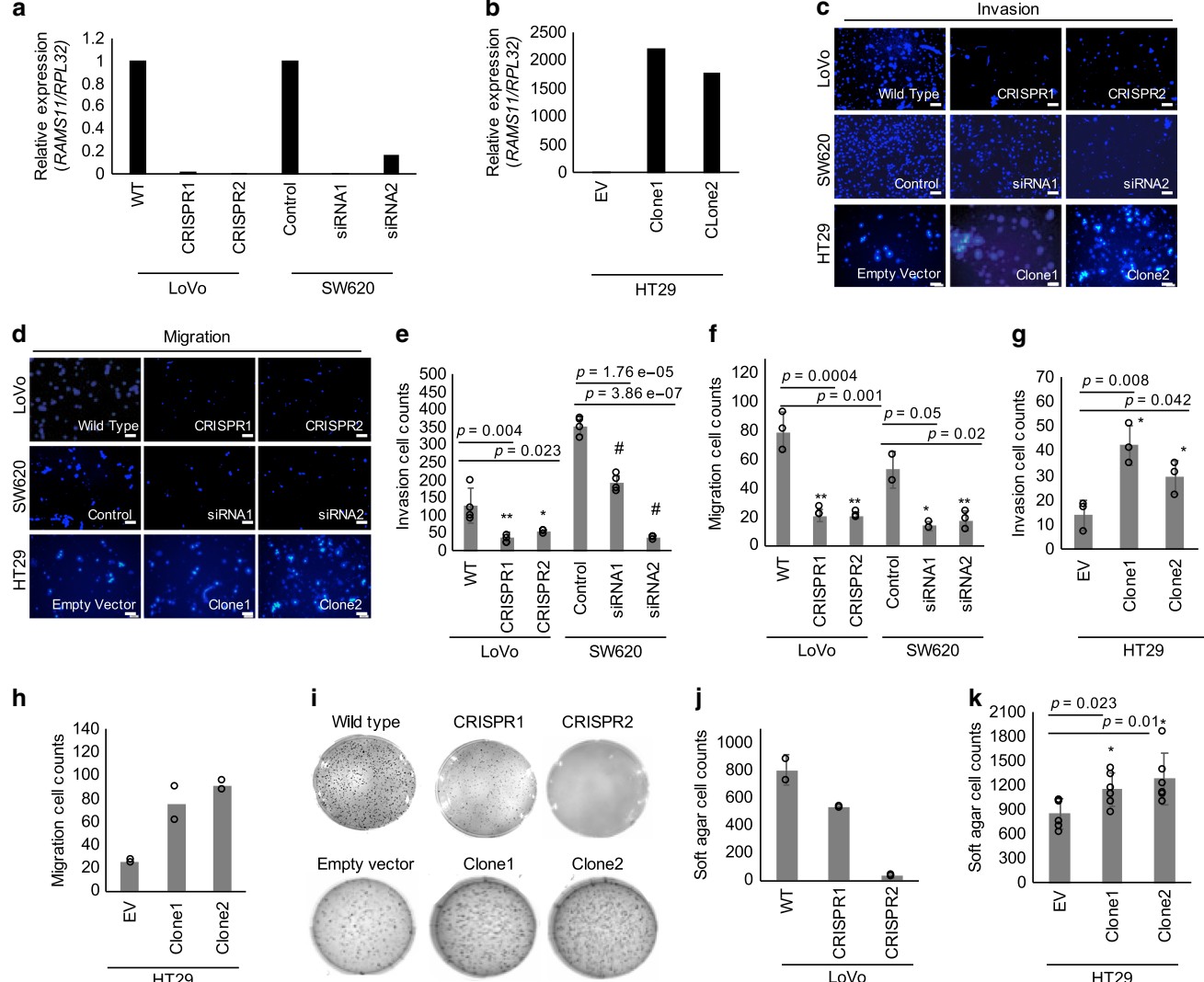

**Fig. 2 RAMS11 promotes an invasive phenotype.** Expression of *RAMS11* in **a** LoVo CRISPR KO, SW620 silenced cells, and **b** HT29 overexpressing cells (Clone1 and Clone2) as measured by qPCR. **c** Images of DAPI-stained LoVo *RAMS11* CRISPR KO cells and SW620 siRNA silenced cells show decreased invasion compared with controls. HT29 cell lines overexpressing *RAMS11* show increased invasion. **d** Images of LoVo *RAMS11* CRISPR KO cell lines and SW620 cells transfected with *RAMS11* siRNAs show decreased migration. HT29 cell lines overexpressing *RAMS11* show increased migration. **e, f** Quantification of invaded ($n = 4$) and migrated cells ($n = 3$) in LoVo and SW620 cells. **g, h** Quantification of invaded ($n = 3$) and migrated cells ($n = 2$) in HT29 cells. **i** *RAMS11* CRISPR KO cells decreased growth on soft agar ($n = 2$) and *RAMS11* overexpressing cells ($n = 6$) increased growth on soft agar. **j, k** Quantification of soft agar cells. All data are presented as mean values ± s.d, analyzed by two-tailed paired *t*-test, and repeated more than two times. Bar = 25 μM, *$p < 0.05$, **$p < 0.005$, #$p < 0.0005$. Source data are provided as a Source Data File.

knockdown of *RAMS11* in a second colon cancer metastatic cell line, SW620, and observed at least 80% knockdown in two independent siRNAs (Fig. 2a). We saw a 50% decrease of invaded cells relative to control cells that were transfected with scrambled siRNA ($p < 0.00005$, two-tailed paired *t*-test, Fig. 2c, e). Conversely, stably overexpressing *RAMS11* (Clone1 and Clone2) in HT29 cells, with low endogenous *RAMS11* expression (Fig. 2b), resulted in a 53% increase in cellular invasion (Clone1 $p = 0.008$, Clone2 $p = 0.042$, two-tailed paired *t*-test) relative to the empty vector control cell line (Fig. 2c, g). Further, we rescued the number of invaded cells to wild-type levels with transient overexpression of *RAMS11*. Our CRISPR KO models re-expressing *RAMS11* (CRISPR *RAMS11* OE) revealed more than a 60% increase of invaded cells relative to the CRISPR KO cell lines (CRISPR1 *RAMS11* OE and CRISPR2 *RAMS11* OE $p < 0.00005$, two-tailed paired *t*-test) (Supplementary Fig. 2d, e). We also observed a 73% decrease in cellular migration in the

CRISPR KO cells (CRISPR1 and CRISPR2 $p < 0.0005$, two-tailed paired *t*-test) and more than 67% decrease in SW620 *RAMS11* silenced cells ($p < 0.05$, two-tailed paired *t*-test) (Fig. 2d, f). In addition, there was increased migration in HT29 *RAMS11* overexpressing cells (Fig. 2d, h). Taken together, this demonstrates that *RAMS11* promotes cellular invasion in CRC.

We next investigated the ability of *RAMS11* to promote anchorage-independent growth as another indication of aggressive oncogenic phenotypes. Using a soft agar colony formation assay, we detected more than a 66% reduction in colony formation in the *RAMS11* CRISPR KO cell lines relative to the wild-type LoVo cell line (Fig. 2i, j). Conversely, there was a 30% increase in colony formation in the *RAMS11* overexpressing HT29 cell lines (Clone1 and Clone2 $p < 0.05$, two-tailed paired *t*-test) compared with the empty vector cell line (Fig. 2i, k). These combined data show that decreased expression of *RAMS11* in genetically modified and transient knockdown cell lines mitigates

aggressive phenotypes, while overexpressing *RAMS11* promotes aggressive phenotypes.

As another hallmark of aggressive phenotypes, we next assessed the effect of *RAMS11* expression on cellular proliferation. We observed a 27% decrease in proliferation in our CRISPR KO cell lines ($p < 0.05$, two-tailed paired $t$-test) (Supplementary Figs. 2f, g, and 10). Taken together, our in vitro data demonstrate that *RAMS11* can promote multiple oncogenic phenotypes.

Next, we evaluated whether *RAMS11* is broadly deregulated across cancer types, which would suggest a critical conserved oncogenic role in cancer progression, which we refer to as an onco-lncRNA[37]. We conducted a pan-cancer analysis of 6984 tissues comprised of matched and unmatched normal and primary tumors across 22 different cancer types studied within the TCGA. This analysis revealed that *RAMS11* had elevated expression in primary tumors compared with normal tissue of origin in colorectal adenocarcinoma ($p < 0.00001$) and four additional cancer types including: lung adenocarcinoma ($p < 0.00001$), lung squamous cell carcinoma ($p < 0.00001$), head and neck squamous cell carcinoma ($p < 0.00001$), and kidney renal papillary cell carcinoma ($p < 0.00001$) (Supplementary Fig. 3a).

Last, since we found that *RAMS11* is an onco-lncRNA upregulated across cancer types, we determined if *RAMS11* also promoted oncogenic phenotypes in additional cancer types. Therefore, we silenced *RAMS11* expression and assessed invasion in two different histologies of non-small cell lung cancer, lung squamous (HCC95) and lung adenocarcinoma (A549), cell line models. We found that silencing *RAMS11* expression in both cancer cell lines caused a decrease in cellular invasion (HCC95 $p < 0.05$; A549 $p < 0.005$, two-tailed paired $t$-test) (Supplementary Fig. 3b–g). These results indicate that increased *RAMS11* expression promotes oncogenic phenotypes in multiple cancer types.

**RAMS11 promotes tumor growth and metastasis in vivo**. Since our *RAMS11* CRISPR KO lines had a significant decrease in cellular proliferation in vitro (Supplementary Fig. 2f, g), we evaluated tumor growth by injecting *RAMS11* CRISPR KO cells into NOD/SCID immunocompromised mice. Twenty-five days after subcutaneous injection of LoVo luciferase-tagged wild-type and our luciferase-tagged *RAMS11* CRISPR KO cells we found a significant decrease ($p < 0.0005$, two-tailed paired $t$-test) in both tumor volume and size in mice injected with *RAMS11* CRISPR KO cells compared with wild-type cells (Fig. 3a–d). These results indicate that *RAMS11* may indeed induce tumor formation and promote oncogenesis in vivo.

Next, to assess the contribution of *RAMS11* to cause metastasis in vivo, we used two mouse models of metastasis: a tail vein injection model to study the development of lung metastases and a hemisplenectomy model to study the development of liver metastases. For the tail vein model, we injected LoVo luciferase-tagged wild-type and our luciferase-tagged *RAMS11* CRISPR KO cells into the tail vein of 5-week-old NOD/SCID mice. We monitored the mice at day 0, within 30 min to 1 h post injection, and weekly for metastasis formation with bioluminescence imaging (BLI). All mice were injected successfully showing similar luminescence levels determined by BLI at baseline day 0 (Fig. 4a). Further, images from day 7 showed no detectable signal indicating the internalization of circulating cells throughout the mouse. There was little or no lung metastasis in mice injected with *RAMS11* CRISPR KO cell lines by day 35 ($p = 0.02$, two-tailed paired $t$-test) as compared with wild-type cells (Fig. 4a, b). We continued to monitor lung metastasis for 91 days and saw significantly less lung metastasis in *RAMS11* CRISPR KO cell-injected mice compared with wild-type cell-injected mice

($p < 0.05$, two-tailed paired $t$-test) (Fig. 4b, c). We also detected less lung metastasis ex vivo in mice injected with *RAMS11* CRISPR KO cells compared with mice injected with wild-type cells (Fig. 4d). In addition, extracted lungs had little to no tumors detected by hematoxylin and eosin (H&E) stain and lower levels of Ki67 staining from *RAMS11* CRISPR KO cell-injected mice compared with wild-type cell-injected mice (Fig. 4e).

We assessed if *RAMS11* promoted liver metastasis using the hemisplenectomy model. LoVo luciferase-tagged wild-type and luciferase-tagged *RAMS11* CRISPR KO cells were injected into 8-week-old NGS mouse spleens[38]. We detected liver metastasis by day 7 in mice injected with wild-type cells (Fig. 5a) and detected significantly lower levels of bioluminescence in *RAMS11* CRISPR KO cell-injected mice compared with wild-type cell-injected mice by Day 21 (CRISPR1 $p = 0.016$, CRISPR2 $p = 0.008$, two-tailed paired $t$-test, Fig. 5a, b). We excised all mouse livers and validated the decrease of liver metastasis (Fig. 5c), decreased liver weights (CRISPR1 $p = 0.0000026$, CRISPR2 $p = 0.00069$, two-tailed paired $t$-test, Fig. 5d), and decrease in overall liver metastasis area (CRISPR1 $p = 0.00021$, CRISPR2 $p = 0.00018$, two-tailed paired $t$-test, Fig. 5e) in *RAMS11* CRISPR KO cell-injected tumors. Decreased tumor burden and proliferation in *RAMS11* CRISPR KO cell livers were further determined by H&E and Ki67 staining (Fig. 5f). Overall, our cell models manipulating *RAMS11* expression demonstrate the ability of *RAMS11* to promote invasive phenotypes both in vitro and in vivo.

**Drug screen reveals RAMS11 resistance to TOP2α inhibitors**. To implicate *RAMS11* in specific biological processes and establish its clinical importance, we conducted a high-throughput viability assay using 119 FDA-approved anticancer drugs from the NIH Developmental Therapeutics Program (Approved Oncology Drugs Set VI). The FDA-approved anticancer panel included multiple classes of drugs such as kinase inhibitors, alkylating agents, antineoplastic antibiotics, anthracycline antibiotics, and antineoplastic agents (topoisomerase inhibitors). The HT29 *RAMS11* overexpressing and control cells were treated for 72 h to assess cellular viability upon drug treatment (Supplementary Fig. 4a and Supplementary Data 3). The *RAMS11* overexpressing cells were resistant to nine drugs as demonstrated by a greater than three-fold increase in cellular viability when compared with the empty vector control cell line with the greatest resistance observed with gemcitabine and floxuridine (FUDR) (Supplementary Fig. 4b, c). FUDR, a 5-FU derivative, is commonly used to treat mCRC, while gemcitabine is used in refractory mCRC[39–42]. Due to 5-FU commonly used to treat mCRC, we further determined if *RAMS11* expression altered drug sensitivity in treated cells. In our *RAMS11* CRISPR KO lines we found a 1.7-fold and 5.8-fold increase in drug sensitivity in CRISPR1 and CRISPR2, respectively, compared with wild-type cells (Supplementary Fig. 5a). Similarly in SW620 cells with transient silencing of *RAMS11* there was a greater than 1.5-fold increase in drug sensitivity in both siRNAs (siRNA1 fold > 1.53, siRNA2 fold > 1.59) relative to scrambled control wild-type treated cells (Supplementary Fig. 5b). 5-FU, irinotecan (topoisomerase I inhibitor (*TOP1*)), and oxaliplatin (new-generation platinum compound) are currently used as first-line active chemotherapy options individually or in combination for patients with metastatic disease[4,43,44]. We did not see a significant effect of cell viability for irinotecan or oxaliplatin using our HT29 *RAMS11* overexpressing cells or LoVo *RAMS11* CRISPR KO cells (Supplementary Data 3, Supplementary Fig 5c–e). SW620 cells with silenced *RAMS11* also did not have a significant effect of cellular viability for oxaliplatin treatment, but we did detect an increase in

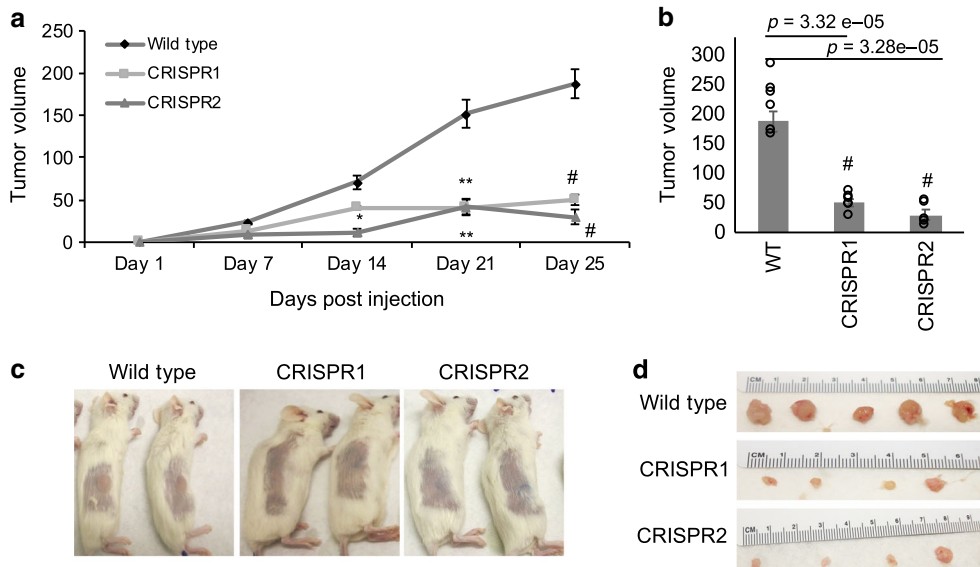

**Fig. 3 RAMS11 induces tumor growth in vivo. a** Significant decrease in tumor growth in *RAMS11* CRISPR KO subcutaneous injected mice. Day 14 CRISPR1 *p* = 0.01 CRISPR2 *p* = 0.0002, day 21 CRISPR1 *p* = 0.00002 CRISPR2 *p* = 0.0006, day 25 CRISPR1 *p* = 3.32e−05 CRISPR2 *p* = 3.28e−05. **b** Quantification at day 25 showing decreased tumor growth in *RAMS11* CRISPR KO lines compared with wild type. **c** Representative mice showing little to no tumor growth and **d** representative resected tumors from mice. Data shown as mean ± SEM and analyzed by two-tailed paired *t*-test, with *n* = 10 per group repeated two times. *\*p* < 0.05, *\*\*p* < 0.005, *#p* < 0.0005. Source data are provided as a Source Data File.

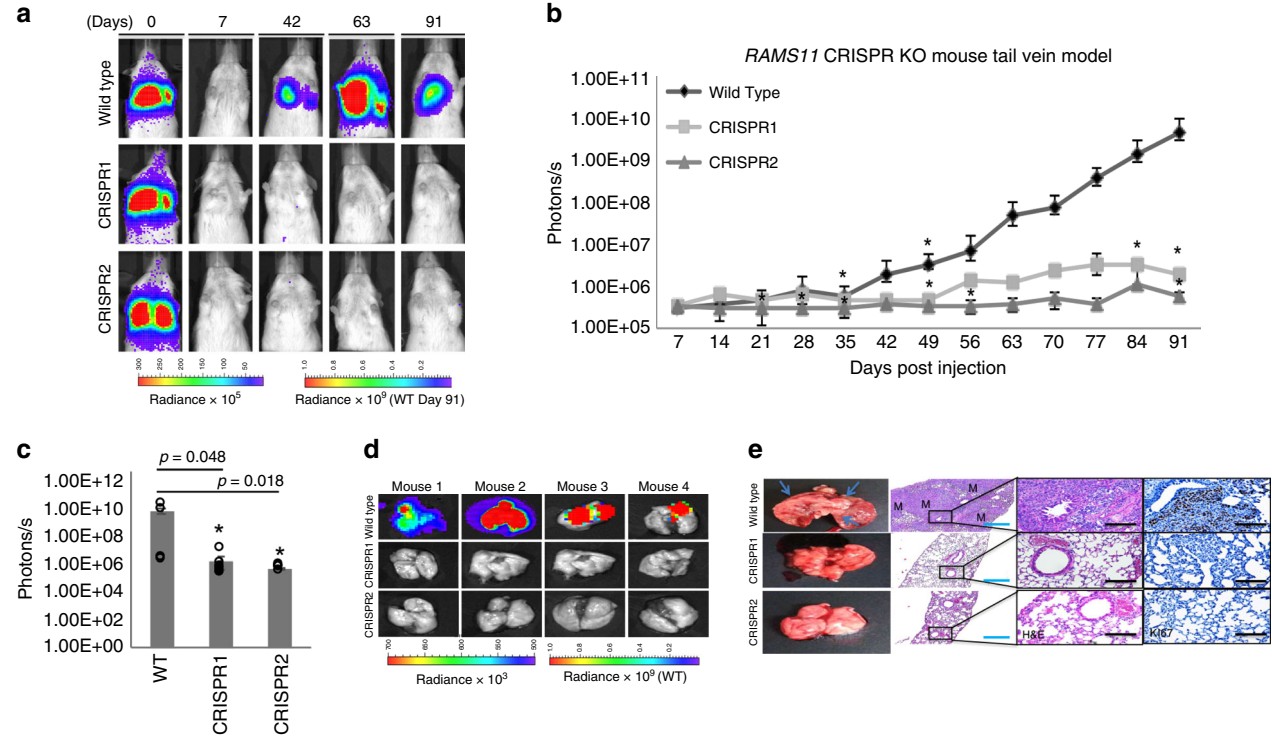

**Fig. 4 RAMS11 induces lung metastasis via tail vein mouse model. a** Representative mice and **b** quantification showing no lung metastasis in *RAMS11* CRISPR KO cell-injected mouse by BLI. Day 28 CRISPR2 *p* = 0.04, day 35 CRISPR1 *p* = 0.02 CRISPR2 *p* = 0.013, day 49 CRISPR1 *p* = 0.02 CRISPR2 *p* = 0.01 day 84 CRISPR1 *p* = 0.05 CRISPR2 *p* = 0.03, day 91 CRISPR1 *p* = 0.04 CRISPR2 *p* = 0.01. **c, d** Day 91 ex vivo mouse lungs show *RAMS11* CRISPR KO cell-injected mice have decreased lung metastasis by BLI. **e** Hematoxylin and eosin stain showing metastasis (M) and Ki67 stain. Three independent tissues were stained per group. Blue bar = 1 mM, black bar = 25 μM. Data shown as mean ± SEM and analyzed by two-tailed paired *t*-test, with *n* = 12 per group repeated two times. *\*p* < 0.05. Source data are provided as a Source Data File.

drug sensitivity to irinotecan (siRNA1 fold > 3.17 and siRNA2 fold > 11.8, Supplementary Fig 5f).

Interestingly, half of the topoisomerase inhibitors assessed caused at least a two-fold increase in drug resistance in the HT29

*RAMS11* overexpressing cells including the TOP1 topotecan hydrochloride (HCl) and four TOP2α inhibitors (doxorubicin HCl, epirubicin HCl, daunorubcin HCl, and idarubicin HCl (Fig. 6a, b). To further support our observation that *RAMS11*

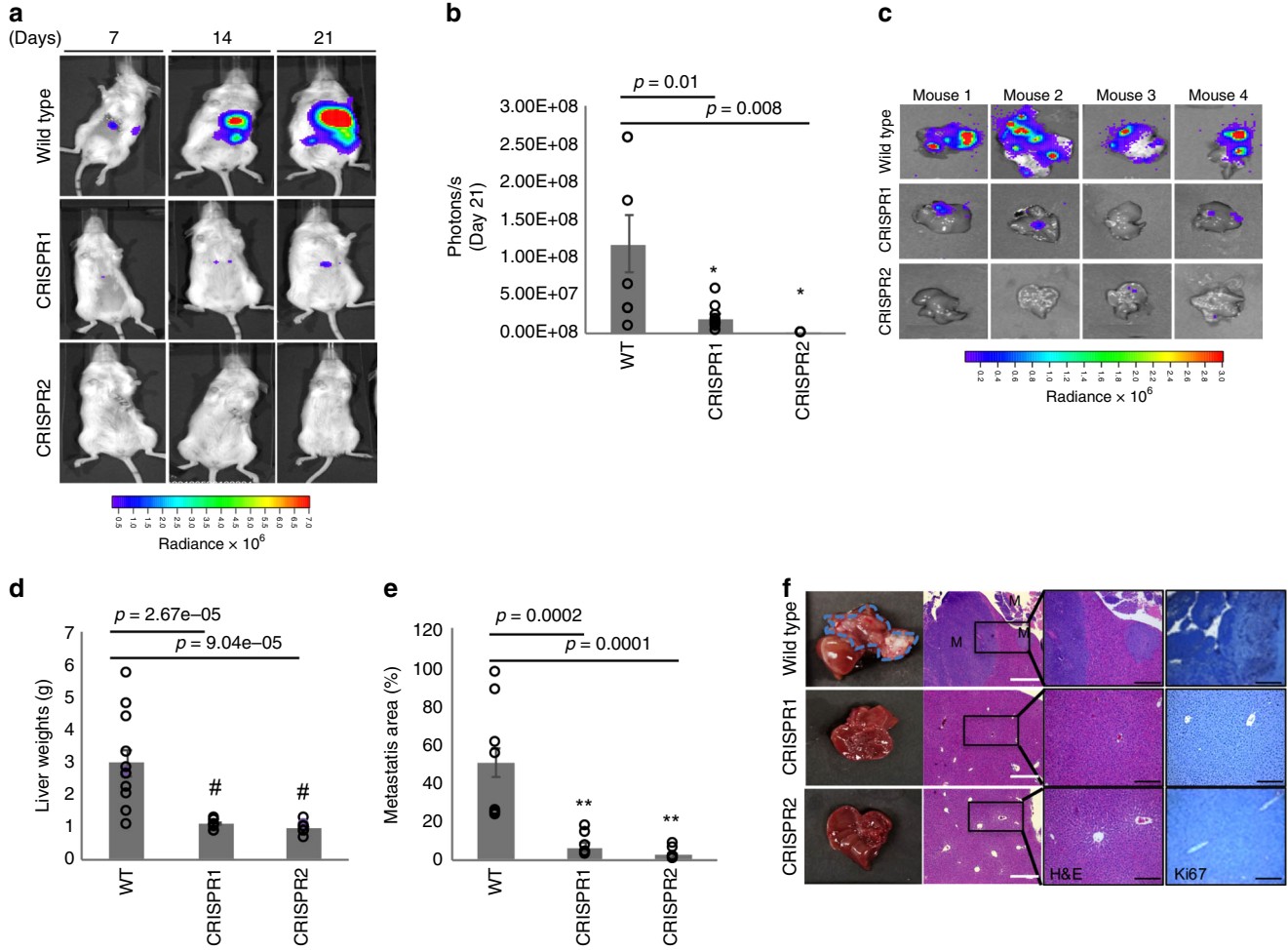

**Fig. 5 RAMS11 induces liver metastasis via hemisplenectomy mouse model. a** Representative mice showing no liver metastasis in *RAMS11* CRISPR KO cell-injected mice by BLI. **b** *RAMS11* CRISPR KO cell-injected mice show a significant decrease in liver metastasis by day 21. **c** Day 21 ex vivo mouse livers show decreased metastasis in *RAMS11* CRISPR KO cell-injected mice by BLI. Wild-type cell-injected mice had **d** increased liver weights and **e** liver metastasis compared with CRISPR KO cell-injected mice. **f** Hematoxylin and eosin stain of livers showing metastasis (M) and levels of Ki67 stain. Three independent tissues were stained per group. White bar = 10 μM, black bar = 100 μM. Data shown as mean ± SEM and analyzed by two-tailed paired *t*-test, with WT *n* = 18, CRISPR1 *n* = 11, CRISPR2 *n* = 11 per group, experiment was repeated three times. *$p < 0.05$, **$p < 0.005$, #$p < 0.0005$. Source data are provided as a Source Data File.

overexpression promoted resistance to topoisomerase inhibitors, we narrowed our focus on clinically relevant drugs that selectively target the DNA topoisomerase TOP2α, doxorubicin and epirubicin. Measuring cellular viability in LoVo *RAMS11* CRISPR KO cells we showed a 1.5-fold increase in drug sensitivity with 0.5 μM doxorubicin or 0.7 μM epirubicin treatment compared with wild-type treated cells ($p < 0.05$, two-tailed paired *t*-test, Fig. 6c).

To further support our drug panel findings, we evaluated whether *RAMS11* regulated TOP2α protein expression. We observed that *RAMS11* overexpressing cell lines had elevated TOP2α protein expression, whereas our CRISPR KO cells displayed a decrease in TOP2α protein levels (Fig. 6d). The decrease in TOP2α expression in our CRISPR KO cells was rescued by re-introducing *RAMS11* expression in these cells (Fig. 6d). Transient silencing of *RAMS11* in SW620 cells also decreased TOP2α protein and mRNA levels relative to our scrambled control (Supplementary Fig. 6a, b). To demonstrate the specificity of *RAMS11* regulation of *TOP2α*, we confirmed there was only a decrease of *TOP2α* mRNA expression by making primers specifically targeting *TOP2α* and not *TOP2β* in the LoVo CRISPR KO and SW620 silenced cell lines

(Supplementary Fig. 6b, c). Lastly, we assessed downstream targets of TOP2α in our CRISPR KO cell lines and observed a decrease in *MLH1* and *ERCC2* mRNA levels supporting *RAMS11* regulation of *TOP2α* (Supplementary Fig. 6d). Overall, our high-throughput drug panel established that *RAMS11* expression impacted cellular sensitivity to topoisomerase inhibitors, specifically inhibitors targeting TOP2α, which led to the discovery that *RAMS11* overexpression increased TOP2α protein expression in colon cancer cell lines. These data highlight the potential for *RAMS11* expression to serve as an important biomarker to select mCRC patients that may potentially benefit from topoisomerase inhibitor treatment.

**RAMS11 binds to chromobox protein 4 (CBX4) to regulate TOP2α.** Due to the nuclear localization of *RAMS11* we hypothesized that it may transcriptionally regulate *TOP2α* expression to increase protein levels and promote topoisomerase resistance. Notably, a recent study found that CBX4 bound to the promoter of *Top2α*[45]. Since CBX4 is known to possess both activation and repressive activities[46,47], and has been found to interact with lncRNAs[47,48], we hypothesized that it could interact with *RAMS11*

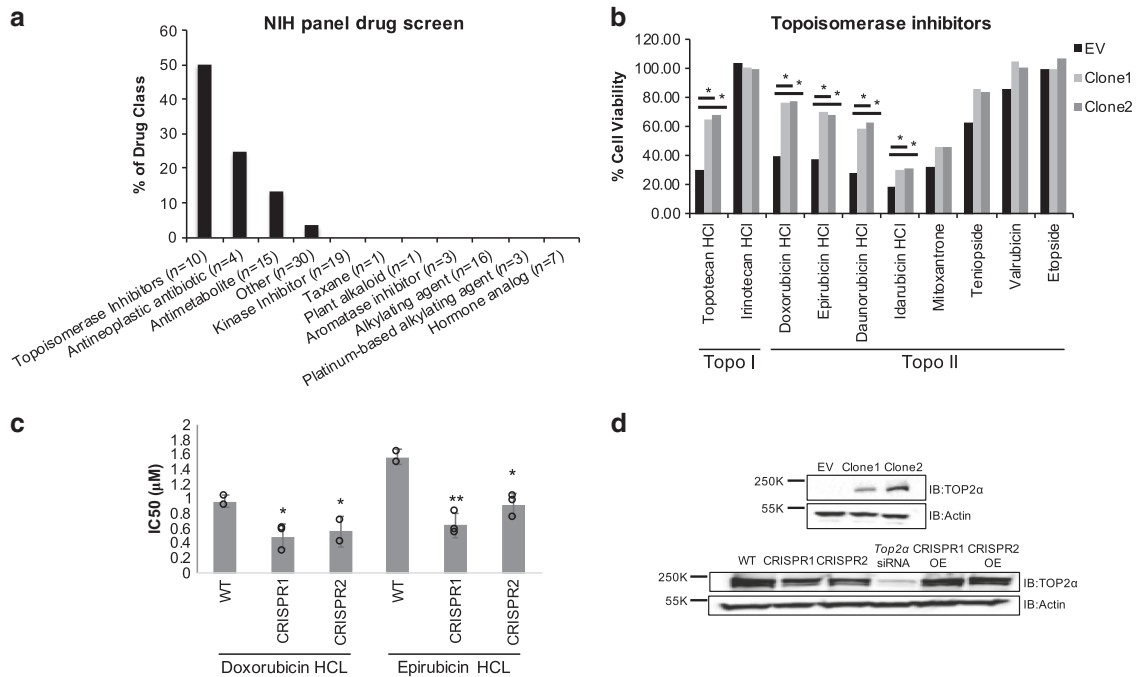

**Fig. 6 *RAMS11* expression alters sensitivity to topoisomerase inhibitors. a** Cell viability assay comparing HT29 empty vector cells with *RAMS11* overexpressing cell lines showing significant resistance to various drug classes. **b** *RAMS11* overexpressing cells have increased cell viability compared with empty vector cells in five of ten topoisomerase (Topo) inhibitors. **c** IC50 values of *RAMS11* CRISPR KO cell lines ($n = 3$) with decreased viability to doxorubicin hydrochloride (HCl) and epirubicin HCl drug treatments. **d** Protein expression of Top2α in (top) *RAMS11* overexpressing cell lines and (bottom) CRISPR KO cell lines, TOP2α siRNA control cells, and CRISPR cell lines overexpressing (OE) *RAMS11*. Band intensities were quantified from the digital image in ImageJ and are shown normalized to the empty vector or wild-type lane for each target. Samples derived from the same experiment and blots were processed in parallel. All data are presented as mean values ± s.d. Experiments repeated three times. *Fold change >1.5. Source data are provided as a Source Data File.

and transcriptionally regulate *TOP2α* expression through interaction with CBX4. We found an 83-fold and 16-fold enrichment of *RAMS11* bound to CBX4 in LoVo and SW620 cells, respectively, as determined by an RNA immunoprecipitation (RIP) coupled with qPCR (Fig. 7a, b). To orthogonally validate these findings, we conducted a RNA pull-down assay utilizing a 5′ Bromo-UTP full-length *RAMS11* sense labeled probe and a negative control anti-sense probe to pull-down proteins that may be bound to *RAMS11*. We found that the *RAMS11* sense probe was bound to CBX4 protein compared with the antisense probe (Fig. 7c) by Western blot of nuclear lysates. In order to identify the regions of *RAMS11* that bind to CBX4, we conducted in vitro RNA pull down in the LoVo cell line using four truncated *RAMS11* fragments (Supplementary Fig. 7a). We re-validated our previous findings that full-length *RAMS11* binds to CBX4 and revealed that nucleotides 600–959 of *RAMS11* interact with CBX4 protein (Supplementary Fig. 7b). These orthogonal methods support *RAMS11* binding to CBX4 protein.

We next evaluated if CBX4 interacts with the *TOP2α* promoter and whether this was dependent on *RAMS11* expression. Binding of CBX4 to the promoter of *TOP2α* was confirmed by chromatin immunoprecipitation (ChIP) coupled with qPCR in LoVo colon cancer cells. Silencing CBX4 led to a 78% decrease in CBX4 occupancy at the *TOP2α* promoter in LoVo cells ($p = 0.021$, two-tailed paired *t*-test, Fig. 7d). Further, we demonstrated this binding is dependent on *RAMS11* expression since there was a greater than 68% decrease in CBX4 occupancy in the *TOP2α* promoter in our *RAMS11* CRISPR KO models ($p < 0.005$, two-tailed paired *t*-test, Fig. 7d). We also observed a decrease in tri-methylation of lysine 4 on the Histone H3 protein subunit (H3K4me3), a modification commonly associated with active transcription, in CBX4 siRNA treated cells and our *RAMS11*

CRISPR KO models (CRISPR1 $p = 0.001$ and CRISPR2 $p = 0.0008$, two-tailed paired *t*-test, Fig. 7e). Decreased CBX4 occupancy and H3K4me3 at the *TOP2α* promoter was further confirmed in the SW620 cell line with transiently silenced CBX4 or *RAMS11* (CBX4 $p = 0.004$, *RAMS11* siRNA $p < 0.05$, two-tailed paired *t*-test, Fig. 7f, g). In addition to demonstrating endogenous binding of *RAMS11* to CBX4 in LoVo and SW620 cells, we found more than 15000-fold enrichment of CBX4 binding to *RAMS11* in our HT29 *RAMS11* overexpressing cells compared with empty vector (Fig. 7h). In addition, the HT29 *RAMS11* overexpressing cells had increased occupancy of CBX4 ($p = 0.0005$ and $p = 0.004$, two-tailed paired *t*-test) and increased occupancy of H3K4me3 ($p = 0.01$ and $p = 0.001$, two-tailed paired *t*-test) at the *TOP2α* promoter (Fig. 7i, j). Further, we rescued CBX4 (Fig. 7k) and H3K4me3 ($p = 0.005$, two-tailed paired *t*-test, Fig. 7l) occupancy by re-introducing *RAMS11* expression into the LoVo CRISPR KO cells. The decrease in TOP2α protein expression was confirmed by Western blot in *RAMS11* CRISPR KO and *CBX4* silenced cells (Fig. 7m). We also observed a decrease in cellular invasion when *CBX4* or *TOP2α* were transiently silenced in LoVo cell lines (Supplementary Fig. 8a, b). Collectively, these data demonstrate *RAMS11*-dependent CBX4 binding to the *TOP2α* promoter. Taken together, we provide evidence of *RAMS11*-dependent CBX4 regulation of *TOP2α* to induce the metastatic phenotype in CRC (Fig. 8).

## Discussion
In the current study we performed transcriptome sequencing of matched normal, primary, and distant metastatic patient samples to identify lncRNAs associated with metastatic progression that

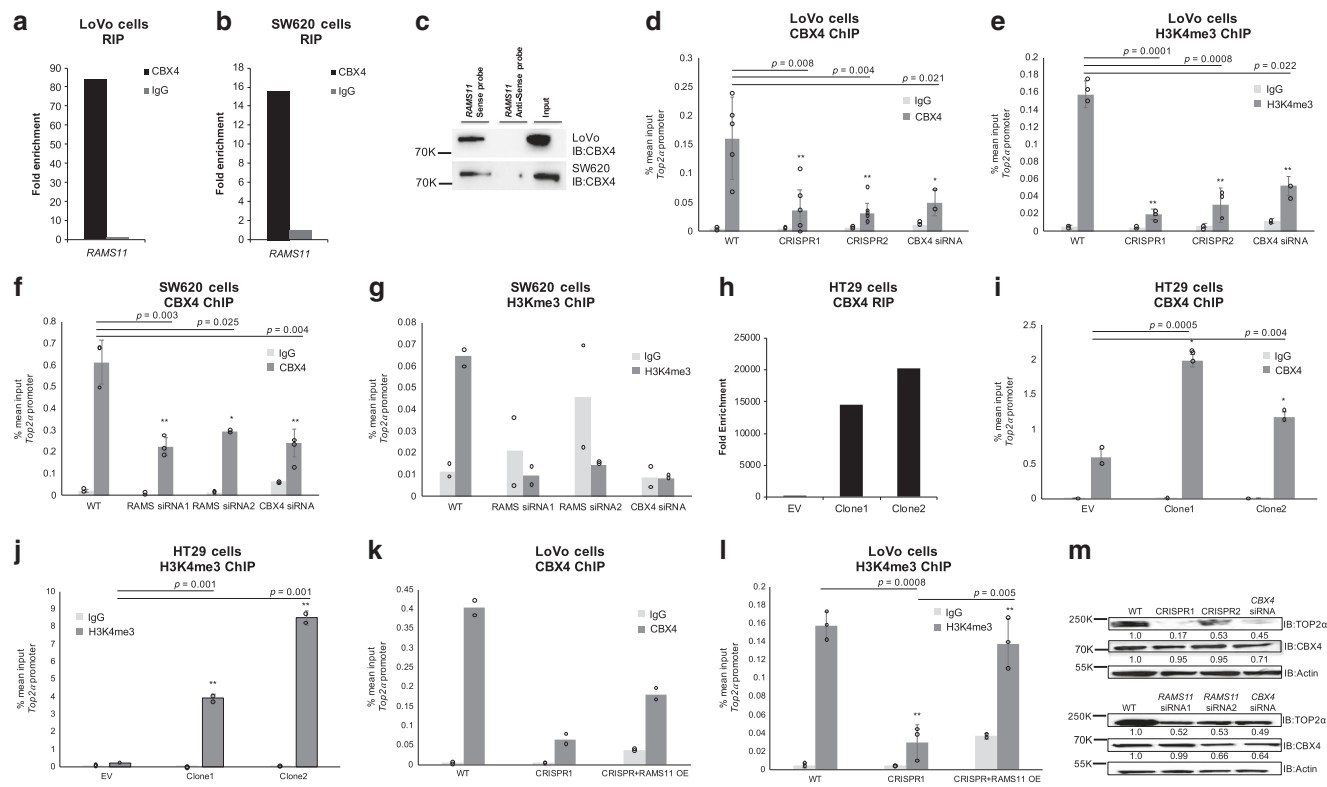

**Fig. 7 RAMS11 binds to chromobox 4 (CBX4) to regulate expression of Top2α mRNA and protein.** RNA immunoprecipitation (RIP) shows binding of *RAMS11* to CBX4 and not negative control IgG in **a** LoVo and **b** SW620 cells. **c** RNA pull down of 5-Bromo-UTP full-length *RAMS11* probe showing binding of CBX4 by Western blot in LoVo and SW620 cells. **d–g** Decreased binding of CBX4 and active histone mark H3K4me3 at *TOP2α* promoter with silenced *RAMS11* expression in chromatin immunoprecipitation (ChIP) assay. IgG $n = 2$, CBX4 $n > 3$, H3K4me3 $n > 2$. **h** RIP showing increased binding of *RAMS11* to CBX4 in HT29 *RAMS11* overexpressing cells. **i, j** ChIP of CBX4 and H3K4me3 shows increased binding to *TOP2α* promoter in HT29 *RAMS11* overexpressing cells. IgG $n = 3$, CBX4 $n = 3$, H3K4me3 $n = 2$. **k, l** ChIP of CBX4 and H3K4me3 in CRISPR KO cells with *RAMS11* overexpression (OE) rescue at *TOP2α* promoter. IgG $n = 2$, CBX4 $n = 2$, H3K4me3 $n = 3$. **m** Protein expression of TOP2α and CBX4 in LoVo (top) and SW620 cell lines (bottom). Band intensities were quantified from the digital image in ImageJ and are shown normalized to the wild-type lane for each target. Samples derived from the same experiment and blots were processed in parallel. Fold change normalized expression to actin is shown below gel. All data are presented as mean values ± s.d, analyzed by two-tailed paired *t*-test. Experiments repeated more than two times. *$p < 0.05$ **$p > 0.005$, #$p < 0.0005$. Source data are provided as a Source Data File.

could serve as a resource for further functional characterization and biomarker studies. To exemplify this, we prioritized *RAMS11* since it was overexpressed in primary and mCRC tumors and its expression correlated with poor disease-free survival. This demonstrates the potential utility of *RAMS11* expression as a marker to stratify high-risk patients. Supporting our clinical findings, we were able to confirm that *RAMS11* promoted oncogenic phenotypes in vitro and in vivo in several cancer types.

To assess the clinical potential of *RAMS11* and elucidate its regulatory mechanism for promoting aggressive phenotypes we used a high-throughput drug assay. We found that antimetabolites gemcitabine and floxuridine had the most significant increase in cellular viability when *RAMS11* was overexpressed. We also found that *RAMS11* promoted resistance to more than half of the topoisomerase inhibitors screened. Currently, the elevated expression of *TOP2α* in primary and mCRC patients[49–51] has served as the rationale for using anthracylines to treat select patients with mCRC. This can be exemplified by an ongoing phase II study to investigate the efficacy of epirubicin as a second-line treatment for patients with *TOP2α* gene amplification and oxaliplatin-refractory mCRC[52] (EudraCT 2013-001648-79). Our study provides mechanistic insight into *RAMS11*-dependent *TOP2α* regulation in mCRC to promote resistance to these inhibitors. In addition, despite the promise of using anthracylines as a mCRC treatment, there are still

many limitations including dose-limiting toxicities, intestinal toxicities, cumulative cardiotoxicity, and off-target effects on *TOP2β* leading to *TOP2β* poisoning[53]. Fortuitously, *RAMS11* specifically targets *TOP2α*, and could be investigated for its therapeutic potential given the increased use of RNA therapeutics, such as locked nucleic acids, in clinical trials.

Currently, several topoisomerase inhibitors are currently FDA approved for treating multiple cancer types and are first-line therapies for breast cancer, bone and soft tissue sarcoma, bladder cancer, anaplastic thyroid cancer, Hodgkin's and non-Hodgkin's lymphoma, and multiple myeloma[53–55]. In addition, *TOP2α* is used as a proliferation marker in multiple cancer types, including CRC[56,57], and elevated levels of *TOP2α* expression are associated with metastasis in prostate cancer, pancreatic cancer, and breast cancer[58–61]. Therefore, the clinical impact of our study extends beyond mCRC and affects a broader patient population given the widespread use of FDA-approved topoisomerase inhibitors coupled with the altered expression of *RAMS11* across multiple solid tumors.

Overall, our understanding of how lncRNAs promote metastasis in CRC patients may have tremendous biological and clinical significance. To address this, our study used patient samples to characterize the landscape of lncRNA expression throughout the progression of primary to mCRC. We also show that lncRNA

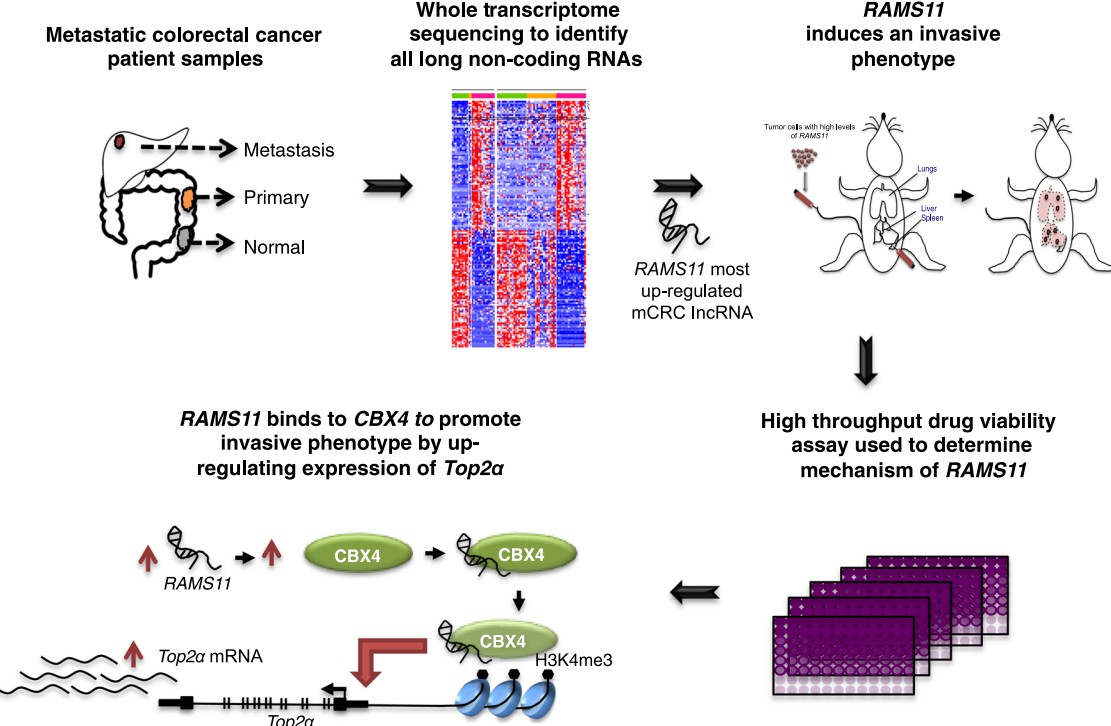

**Fig. 8 RAMS11 identification and model in metastatic colorectal cancer.** Process of identifying *RAMS11* and model showing *RAMS11 CBX4* complex binding to *Top2α* promoter to increase metastatic phenotype.

*RAMS11* directly affects mCRC biology, including promoting an aggressive phenotype and correlating with treatment response and resistance.

## Methods

**Patient samples and RNA sequencing.** Patients were enrolled at Washington University School of Medicine in St. Louis and informed consent was obtained under an IRB-approved protocol. Adjacent normal, primary, and liver metastasis tissues were resected from mCRC patients and fresh frozen prior to RNA extraction (Supplementary Data 4). PolyA cDNA libraries were constructed using NuGen Ovation Kit V2, and paired-end sequencing was performed on Illumina HiSeq 2000. RNA-Seq data from Kim cohort was downloaded from NCBI GEO (GSE50760). TCGA RNA-Seq pre-aligned bam files were downloaded from the Cancer Genomics Hub (http://cghub.ucsc.edu/).

**RNA-Seq data analysis.** The human reference genome assembly version GRCh38/hg38 and the corresponding gene annotations were used in RNA-Seq analysis. Gene annotations were combined from Gencode v23[62], RefSeq downloaded from the UCSC Genome Browser[63], and the Broad lncRNA catalog[64]. Redundant transcripts were removed and overlapping transcripts were assigned to the same gene. RNA-Seq reads were aligned to the human genome using Tophat 2.0.8[65]. Transcript assemblies were generated using Cufflinks 2.1.1[66]. FeatureCounts v1.5.0[67] was used to generate fragment counts for individual transcripts requiring a mapping quality score ≥ 1. FPKM was calculated using transcript fragment counts. For DE analysis, the transcript with highest FPKM among isoforms were selected to represent a gene locus, similar to our previous approach[68]. EdgeR 3.8.6[69] was used to perform a TMM normalization and DE analysis using the raw fragment counts. In the meta-analysis, DE $p$ values were combined using the Stouffer method[70] and fold change was averaged between WUSTL and Kim cohorts. RAMS were defined as lncRNAs that were not tissue specific and DE between metastasis versus primary tumor and between metastasis versus normal tissue (FC ≥ 2, FDR ≤ 0.05).

**Exon array data analysis.** We repurposed the Affymetrix exon array for lncRNA analysis by realigning the probe set sequences against the human transcript sequences using SeqMap 1.0.12[71] allowing one mismatch. Only probe sets consisting of probes that were uniquely aligned to transcripts from the same gene were retained. Exon array expression was processed and normalized using Affymetrix Power Tool 1.18 (https://www.thermofisher.com).

**Survival analysis.** Survival analysis was performed using the Cox proportional hazard model with R survival package 2.37-7 2014 (https://CRAN.R-project.org/package=survival). The median expression of *RAMS11* within a cohort was used to stratifying patients into low and high *RAMS11* expression groups. Kaplan–Meier curves were plotted using the R survplot package 0.0.7 2014 (http://www.cbs.dtu.dk/~eklund/survplot/).

**Cell culture.** Colon cancer cell lines CCD18-Co and SW480 were a kind gift from Dr David Shalloway at Cornell University. All other colon cell lines (HT29, HT-15, DLD1, SW620, Caco-2, HCT-116, and RKO) were a kind gift from Dr A. Craig Lockhart at Washington University. LoVo cell lines were purchased from ATCC (ATCC CCL-229). HCC95 and A549 cell lines were a kind gift from Dr Lauren Michel and Dr Brian Van Tine, respectively, from Washington University. SW620 cells were grown in DMEM (Invitrogen, Carlsbad, CA) with 10% fetal bovine serum (Sigma, St. Louis, MO), and 1% penicillin/streptomycin (Invitrogen, Carlsbad, CA) complete media. LoVo cells were grown in DMEM/F12 (Invitrogen) with 10% fetal bovine serum, and 1% penicillin/streptomycin complete media. HT29, HT-15, DLD1, and Caco-2 cells were grown in McCoys (Invitrogen) with 10% fetal bovine serum and 1% penicillin/streptomycin, and all other cells were grown in RPMI (Invitrogen) with 10% fetal bovine serum, and 1% penicillin/streptomycin complete media.

**Rapid amplification of cDNA ends (RACE).** 5′ and 3′ RACE was done using the GeneRacer Kit (Invitrogen) according to the manufacturer's instructions. RACE PCR products were obtained with Platinum Taq High Fidelity (Invitrogen) using the GeneRacer primer (supplied) and a gene-specific primer found in Supplementary Data 5. Products were visualized on a 2% agarose gel and purified by gel extraction (Qiagen, Germantown, MD). This product was then cloned into pcr4-TOPO vector (Invitrogen) and grown in TOP10 E. coli. Clones were sequenced with the M13 forward primer at the Protein and Nuclei Acid Chemistry Laboratory at Washington University.

**Generation of RAMS11 silenced and overexpression cells.** LoVo *RAMS11* CRISPR KO cell lines were generated through the Genome Engineering and iPSC center at the Washington University. CRISPR/Cas9 was used to create a genomic deletion of the last four exons of *RAMS11* in LoVo metastatic colon cancer cells (Supplementary Fig. 2a). We used two different cell clones for the described experiments. In addition, we silenced expression of *RAMS11* using custom silencer select RNAs (siRNAs) targeting *RAMS11*, *CBX4*, *TOP2α*, or a negative scrambled control (Invitrogen). siRNA sequences are listed in Supplementary Data 5.

Full-length *RAMS11* transcript was PCR amplified from LoVo cells and cloned into the pCFG5-IEGZ vector (a kind gift from Dr Ron Bose, Washington University). Full-length *RAMS11* inserts were confirmed with Sanger sequencing at

the Protein and Nuclei Acid Chemistry Laboratory at Washington University. Retroviral infection of cancer cells was performed according to Kauri et al.[72]. Briefly, the amyotrophic phoenix cell line was transfected with 10 μg of pCFG5-RAMS11 or empty vector control by calcium phosphate precipitation and incubated for 24 h. Viral supernatants were harvested after an additional 24-h incubation. Virus was added to cells seeded in six-well dishes in the presence of 8 μg/mL polybrene (Sigma). Cells were centrifuged at $300 \times g$ for 90 min and fresh media was added to the plate. After 14 days of Zeocin (Invitrogen) selection cells were used for assays. HT29 colon cells that had low endogenous expression of RAMS11 were infected with virus expressing RAMS11 or empty vector for 48 h and selected with 100 μg/mL Zeocin.

**Quantitative real-time PCR**. Total RNA was isolated for each CRC cell line using Takara Bio NucleoSpin RNA (Takara, Mountain View, CA). Total RNA was then transcribed to cDNA with SuperScript III First strand cDNA system (Invitrogen) and quantified using Fast Sybr Green Master Mix (Invitrogen) as per the manufacturer's protocol. Primer sequences are available in Supplementary Data 5.

**Protein detection by Western blot**. Western blots were conducted by plating 250,000 representative cancer cells in a six-well dish. For transient knockdown experiments, the next day cells were transfected at 6.25–25 nM with two independent custom designed siRNAs or a negative scramble control with Lipofectamine RNAiMax (Invitrogen) for 72 h. Cells were then lysed with Tris lysis buffer (50 mM Tris-HCl, 1% Triton X-100, 131 mM NaCl, 1 mM sodium orthovanadate, 10 mM $Na_4P_2O_7$, 10 mM NaF, 1 mM EDTA), run on NuPAGE 4-12% Bis-Tris gel (Invitrogen) and transferred to nitrocellulose membrane (BioRad, Hercules, CA). Blots were then probed overnight at 4° with respective antibodies including TOP2A, CBX4, and ACTIN, then washed with TBST buffer, and then applied with secondary goat anti-rabbit HRP-linked or goat anti-mouse HRP-linked antibodies (Thermo Fisher, Waltham, MA). Blots were then washed, visualized with Clarity Western ECL Substrate (Bio-Rad), and imaged using the ChemiDoc XRS+ System (Bio-Rad). Band intensities were quantified from the digital image in ImageJ and are shown normalized to the control lane for each target. Raw Western blots are shown in Supplementary Fig. 9. All antibodies and concentrations are listed in Supplementary Data 6.

**RNA immunoprecipitation**. RIP coupled to qPCR assays were conducted by isolating nuclear lysates from ten million LoVo or SW620 cells following the NER-PER Nuclear and Cytoplasmic Extraction Reagent Kit (Thermo Fisher). Nuclear lysates were then incubated overnight with 5 μg CBX4 antibody or IgG antibody isotype control in RIPA wash buffer (50 mM Tris-HCl pH 7.4, 150 mM NaCl, 1 mM $MgCl_2$, 1% NP40, 0.5% Na-deoxycholate, 0.05% SDS, 1 mM EDTA) and SUPERase-in RNAse inhibitor (Invitrogen). The next day 50 μL of Invitrogen Dynabeads Protein G were added to the antibody lysate/mixture and were rotated for 1 h at 4°. Next, beads were subsequently washed six times with RIPA wash buffer using a magnetic bead separator. Protein was then digested with proteinase K buffer (RIPA buffer, 10% SDS, 10 mg/mL proteinase K), at 55° for 30 min shaking. RNA was phenol:chloroform:isoamyl alcohol extracted following the general protocol (Thermo Fisher). Last, gDNA was removed from RNA using ArcticZymes Heat and Run gDNA removal kit following the manufacturer's protocol (Tromso, Norway). cDNA was made using SuperScript III First strand cDNA system as indicated above and qPCR was run with Fast Sybr Green Master Mix and indicated primers (Supplementary Data 5). Fold enrichment of qPCR results were calculated following Sigma-Aldrich Data Analysis Calculation Shell by comparing nonspecific control IgG antibody raw CTs to CBX4 RNA binding protein CT normalized against 1% input.

**Chromatin immunoprecipitation**. ChIP qPCR assays were conducted by first sonicating five million cells in SDS lysis buffer (1% SDS, 500 mM EDTA, 50 mM Tris-HCl pH 8). Next, sonicated cells were immunoprecipitated with 5 μg IgG, CBX4, or H3K4me3 antibodies in ChIP dilution buffer (0.01% SDS, 1.10% Triton X-100, 1.2 nM EDTA, 16.7 mM Tris-HCl pH 8, 167 mM NaCl), and 1X Halt Protease and Phosphatase inhibitors overnight with rotation at 4°. The next day Dynabeads Protein G (Invitrogen) were added to the antibody lysate mixture and rotated for 1 h. Bead/lysate mixture was then washed once with low salt wash buffer (0.1% SDS, 1% Triton X-100, 2 mM EDTA, 20 mM Tris-HCl pH 8, 150 mM NaCl), then high salt buffer (0.1% SDS, 1% Triton X-100, 2 mM EDTA, 20 mM Tris-HCl pH 8, 500 mM NaCl), lithium chloride wash buffer (0.25 M lithium chloride, 1% NP40, 1% sodium deoxycholate, 1 mM EDTA, 10 mM Tris-HCl pH 8), and finally two washes with Tris-HCl EDTA buffer (10 mM Tris-HCl pH 8, 1 mM EDTA). DNA was eluted by incubating beads for 30 min at room temperature with SDS elution buffer (1% SDS, 0.1 M sodium bicarbonate), followed by 1.25 M NaCl and 2.5 mg/mL RNAse A at 95° for 15 min shaking followed by addition of proteinase K buffer (1 μL 10 mg/mL proteinase K, 5 μL 0.5 μM EDTA, 10 μL 1 M Tris pH 7.5) shaking at 60° for 15 min. DNA was then isolated using phenol:chloroform:isoamyl alcohol extraction following the general protocol as mentioned above. DNA was diluted by five and used for qPCR. The % input calculations were determined by comparing CT values from input DNA and ChIP DNA for the TOP2A target promoter region using the following equation: %Input = % of starting input

fraction $\times 2^{[CT(input)-CT(ChIP)]}$. Primer sequences are available in Supplementary Data 5.

**BrU-labeled RNA pull down**. Full-length RNA probes and fragmented RAMS11 probes were made using the Promega Riboprobe in vitro transcription kit from 2.5 μg of linearized DNA in the pGEM-3Z vector (Madison, WI). Antisense probes were made by in vitro transcription from the SP6 promoter. RAMS11 RNA pull-down experiments were performed in LoVo and SW620 nuclear lysates following the RiboTrap Kit manufacturer's protocol (MBL, Woburn, MA). Truncated RAMS11 probes consisted of fragments 1–250, 200–450, 400–650, and 600–959 basepairs of RAMS11. RAMS11 probes were synthesized and subcloned at Gene Universal (Newark, DE).

**Nuclear cytoplasmic isolations**. Nuclear and cytoplasmic isolations were conducted using the PARIS Kit (Thermo Fisher) following the manufacturer's protocol. Total RNA was also collected as described above. Nuclear and cytoplasmic isolations were calculated by normalizing respective gene to total RNA expression.

**Transwell assays**. Cell lines were seeded at 300,000 cells in a six-well dish. The next day cells were transfected with siRNAs targeting RAMS11, TOP2α, CBX4, or overexpression plasmids. Seventy-two hours later cells were harvested and reseeded at 200,000 cells on a transwell 8.0 μM permeable membrane support (Corning, Corning, NY) in 24-well plates in a modified Boyden chamber assay. A serum gradient was established with cells plated in serum-free media and complete media (10% FBS) added to the bottom of the well. For invasion assays, transwells were precoated with 200 μg/mL Matrigel (Corning) before addition of cells. Cells were allowed to migrate or invade overnight and then fixed with 4% paraformaldehyde (Electron Microscopy Sciences, Hatfield, PA), and nuclei were stained with DAPI (Sigma, 1 μg/μL). A cotton swab was used to remove cells from the top of the membrane. Migrated DAPI-stained cells were imaged with Q-Capture Pro software on an Olympus IX70 microscope, quantified using ImageJ software (http://imagej.nih.gov/ij/), and statistical significance was determined by a Student's two-tailed $t$-test. Four to seven images were taken per transwell membrane at ×20 magnification. Assays were repeated two to three times.

**Soft agar assays**. HT29 cells overexpressing RAMS11 and LoVo RAMS11 CRISPR KO cell lines were resuspended at 75,000 cells in 0.4% Difco soft agar (BD Bioscience, Franklin Lakes, NJ) and seeded onto a 5% Difco base agar. Cells were given fresh media every 3 days for around 2 weeks, and once colonies were visible by eye, cells were stained with 0.5% Crystal Violet (Sigma) for 3 h. Plates were then imaged using the ChemiDoc XRS+ (Bio-Rad) and counted with ImageJ software. Average cell counts were used for comparison and statistical significance was determined by a Student's two-tailed $t$-test. Assays were repeated three times.

**Drug treatments**. The NIH approved oncology library of 119 drugs (AOD6 plate 4825-1 and AOD6 plate 4826) was received from the NIH National Cancer Institute DTP Developmental Therapeutics Program. Drugs were diluted in DMSO to 1 mM and the well assignments were rearranged so drugs were confined to the inner 60 wells of 96-well plates. HT29 RAMS11 overexpressing Clone1 and Clone2 cell lines were seeded at 5000 cells per well in a 96-well plate. The next day serial diluted drug was added to pre-seeded plates in media containing 1% DMSO vehicle, using a Robbins Hydra 96 microdispenser. Two 96-well plates of cells were used as vehicle controls. The plates were incubated for 3 days. Percent viability was scored by incubating cells for 3 h with resazurin sodium (0.023 mg/mL, Sigma R7017). The reaction was stopped by the addition of SDS (1% final concentration). Fluorescence Ex/Em 540/590 was read in a Biotek Synergy H1 plate reader (Winooski, VT). The fluorescence values for the vehicle plates were averaged and percent viability was determined by the formula: Percent viability = (average vehicle − value)/(average vehicle − average resazurin in media blank) × 100. We removed drugs that were undetectable, or out of range, by resazurin assay, leaving 118 drugs to assess in the study (Supplementary Data 3). Values with more than a 1.5-fold change in both RAMS11 Clone1 and Clone2 overexpressing cell lines were used to determine significance. Individual drug IC50 assays were done in a similar manner as described above with CRISPR KO cells. Assays were repeated more than three times.

**In vivo models**. The animal studies were reviewed and approved by the Washington University's Institutional Animal Care and Use Committee protocol. For subcutaneous injections, 2e6 LoVo wild-type, RAMS11 CRISPR1, or RAMS11 CRISPR2 luciferase-tagged cells were injected subcutaneously in ten NOD/SCID mice per group. Weekly tumor size was determined by caliper measurements comparing length × width × height × 0.5. For post analysis lung tissues and subcutaneous tumors tissues were removed and formalin fixed and paraffin embedded. This experiment was repeated two times.

In the lung metastasis mouse model, 2e6 LoVo wild-type and CRISPR luciferase-tagged cell models were injected into the lateral tail vein of twelve 5-week-old NOD/SCID mice (Jackson Laboratories, Bar Harbor, MA) per group using 30-gauge needles. In weekly intervals mice were imaged with the Olympus

OV100 Small Animal Imaging System (IVIS Spectrum, Caliper, Hopkinton, MA) in conjunction with the Small Animal Imaging Core (SAIC) at Washington University. Mice were imaged for 1 min with sequential 5-s exposures. Luminescence was quantified using the Living Image Software 3.2 (Caliper). All micrometastasis were imaged at day 0 (30 min to 1 h) post operation and weekly for 12 weeks. At the conclusion of the study, mice were sacrificed and examined visually and with bioluminescence for lung metastases in vivo and ex vivo. Lungs were dissected and formalin fixed and paraffin embedded for histological analysis with H&E and Ki67 staining. This experiment was repeated twice.

For the hemisplenectomy mouse model[38], 2e6 LoVo wild-type and CRISPR luciferase-tagged cells in 50 μL of PBS were injected into the spleen of 6–8-week-old NGS mice (Jackson Laboratories) (WT $n = 18$, CRISPR1 $n = 11$, CRISPR2 $n = 11$) using 30-gauge needles during open laparotomy. Cell injections were followed with a 50 μL PBS flush. Incisions were closed with sutures and surgical clips. In weekly intervals mice were imaged with the Olympus OV100 Small Animal Imaging System (IVIS Spectrum) in conjunction with SAIC. Mice were imaged for 10 s to 1 min exposures. Luminescence was quantified using the Living Image Software 3.2 (Caliper). All micrometastasis were imaged at day 7, day 14, and day 21 post operation. At the conclusion of the study, mice were sacrificed and examined visually and by bioluminescence for liver metastases in vivo and ex vivo. Livers were dissected, weighed, and formalin fixed and paraffin embedded for histological analysis with H&E and Ki67 staining. This experiment was repeated three times.

**Reporting summary**. Further information on research design is available in the Nature Research Reporting Summary linked to this article.

## Data availability

The RNA-Seq data generated in this study (WUSTL cohort) have been deposited in the dbGaP database under the accession code phs001722. The Kim et al.'s data[29] referenced during the study are available in a public repository from the NCBI Gene Expression Omnibus under the accession code GSE50760. The source data underlying all figures are provided as a Source Data File. All the other data supporting the findings of this study are available within the article and its Supplementary Information files and from the corresponding author upon reasonable request. A reporting summary for this article is available as a Supplementary Information file.

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

## Acknowledgements

J.S.F. received funding from the Washington University School of Medicine Molecular Oncology Training Grant (T32CA113275). B.A.K., M.K.H., and J.G.G. received funding from the Washington University School of Medicine Surgical Oncology Basic Science and Translational Research Training Program (T32CA009621). R.C.F. and C.A.M. received funding from The Alvin J. Siteman Cancer Center Siteman Investment Program, The Foundation for Barnes-Jewish Hospital Cancer Frontier Fund, the National Cancer Institute Cancer Center Support Grant P30 CA091842, and the Barnard Trust. R.C.F. also received funding from the American Surgical Association Foundation Fellowship, American Cancer Society Institutional Review Grant, the Society of Surgical Oncology James Ewing Foundation Clinical Investigator Award, the Sidney Kimmel Translational Science Scholar Award, and the David Riebel Cancer Research Fund. Funding was also provided for C.A.M. by a Research Scholar Grant (130878-RSG-17-058-01-RMC) from the American Cancer Society and the NIH CTSA Grant #UL1 TR002345. We would like to thank the Washington University's Genome Engineering and iPSC Center for help in the development of the CRISPR cell lines. We thank the Alvin J. Siteman Cancer Center at Washington University School of Medicine and Barnes-Jewish Hospital in St. Louis, MO., for the use of the Siteman Flow Cytometry, which provided flow cytometry service. The Siteman Cancer Center is supported in part by an NCI Cancer Center Support Grant #P30 CA091842. We would also like to thank the SAIC at Washington University for help with imaging mouse models and the Department of Medicine Pulmonary Morphology Core Division of Pulmonary and Critical Care Medicine for histology preparation. We also thank Jacqueline Payton, MD, PhD, for critically reviewing this manuscript.

## Author contributions

J.S.F., R.C.F., and C.A.M. designed the project. J.S.F., H.X.D., N.M.W., A.C.L., R.C.F., and C.A.M. directed experimental studies. H.X.D., C.R.C., and A.E. performed sequencing data analysis. J.L. performed statistical analysis. J.S.F., N.M.W., M.S.S., B.A.K., E.B.R., G.G.L., J.G.G., M.K.H., and C.T. performed experimental studies. J.M. coordinated and processed biospecimens. J.S.F., H.X.D., N.M.W., C.R.C., R.C.F., and C.A.M. interpreted data. S.P.G., E.R.M., R.K.W., T.J.L., and R.C.F. provided project guidance. E.R.M., R.K.W., R.C.F., and C.A.M. provided financial support. J.S.F., H.X.D., N.M.W., and C.A.M. wrote and edited manuscript, which all authors reviewed.

## Competing interests

The authors declare no competing interests.
