## [Peer Review File · Nature Communications]

Reviewers' comments:

Reviewer #1 (Remarks to the Author):

In this study Silva-Fisher et al., identified lncRNAs associated with the development of metastatic colorectal cancer. They focused on one lncRNA that they named RAMS11. For the most part, the data is strong, the findings are novel and interesting, and the manuscript is well-written. The points below highlight some of the weaknesses and concerns that could be addressed. Overall, this is a nice study, the strengths outweigh the weaknesses and the impact of this study is considered high.

Major Comments:

1. Figure 1: The authors mention in the Figure legend about 5'3' RACE but this data appears to be missing. Also, in Figure 1C, are the numbers on the Y-scale (0, 2) no of reads, RPKM, FPKM, etc?
2. The authors need to mention in the text that RAMS11 is LINC01564. Without this, the Ref Seq ID will be buried in Table S1, as it is now. In fact, it looks like LINC01564 is a part of RAMS11? The 5' and 3'RACE data is not there. If it there, the authors should modify the text accordingly because in the current manuscript, it seems that RAMS11 is a novel transcript. It would be important to determine the length of the RAMS11 RNA by Northern blotting and/or RT-PCR for the full-length lncRNA.
3. Does the expression of RAMS11 correlate with patient survival in other cancer types in addition to COAD?
4. Figure 1D: Strangely, the authors called the first exon on RAMS11 the last exon.
5. The order in which some of the data is presented is awkward. For example, would it not make more sense to show the efficacy of RAMS11 CRISPR knockout and RAMS11 overexpression data (Figures 3J and 3K) before the invasion data in Figure 3 to demonstrate that these systems were effectively constructed before they provide data where these systems were used.
6. Along the same lines, they describe the results of Figure 3 before the results of Figure 2B-2G.

7. The data on positive regulation of TOP2a by RAMS11 is very interesting and novel. It fits very well with the functional data in Figure 5. Is there a positive correlation between RAMS11 expression and TOP2a in CRC patients and is there any link to patient survival? If not in CRC, could the authors check this in other cancer types? Could they also check for correlation between RAMS11, TOP2a and a CBX4-regulated transcriptome signature in CRC patients?

8. Additionally, these analysis between RAMS11, TOP2a, some CBX4-regulated genes and H3K4me3 could be done in the panel of CRC cell lines they already have. The data from these experiments could be useful in determining how broadly applicable the proposed model is.

9. Figure 6C: To show the specificity of RAMS11 interaction with CBX4, the authors can probe the immunoblot for a nonspecific protein such as Histone H3. One would expect to see the nonspecific protein in all 3 lanes.

10. What are the regions of the RAMS11 RNA that bind to CBX4? The authors can identify these regions by in vitro RNA pulldowns similar to Figure 6C but using fragments of RAMS11.

11. Experiments demonstrating the effect of RAMS11 expression on CBX4 binding to Top2a promoter may be strengthened by RAMS11 overexpression and rescue experiments in addition to the RMAS11 knockout.

12. The authors should address the apparent decrease in CBX4 expression after RAMS11 knockout in Figures 6H and 6I.

Minor Comments:

1. The decrease in TOP2a expression in LoVo western blot is not very striking (Figure 6H).

2. The title could be more informative – they mainly study only one lncRNA, so it may be better to include that lncRNA in the title.

3. Could include a reference for “response to second line treatment is even less effective that first line” (Introduction, end of first paragraph).

4. Second paragraph introduction: “significance of lncRNAs can be exemplified by their use as diagnostic, prognostic, and predictive biomarkers, and potential use as therapeutic targets.” Change to: “diagnostic, prognostic, predictive biomarkers, and potential use as therapeutic targets”

5. Third paragraph introduction: include reference for “lncRNAs have been shown to promote tumor progression”

6. Page numbers and/or line numbers are missing.

7. Figure 6C: Change Ant- to Anti-

Reviewer #2 (Remarks to the Author):

In the manuscript by Silva-Fisher et al., the authors performed transcriptomic and meta-analysis of lncRNAs expression in two (small) independent cohorts of CRC patients and identified RAMS11 as a novel lncRNA that promotes metastasis/tumorigenesis in CRC and potentially other cancers. They further performed in vitro and in vivo studies to further characterize phenotypes mediated by RAMS11 that could contribute to tumor progression and metastasis and identified RAMS11 as a novel interacting partner of CBX4 which in turns mediate binding of RAMS11-CBX4 complex to the promoter of TOP2 β .

The study is of novelty given the authors’ finding that RAMS11 is prognostic and indicative of resistance to TOP2 β inhibitors.

I have several major concerns:

1. Clinical: The potential clinical impact & relevance is presently assessed to be low and overstated by the authors. This study is not likely in the near to medium term be translatable to the clinic in guiding treatment options in the adjuvant for curative intent patients or in the palliative second/third line treatment options for refractory CRC patients

2. Phenotypic characterisation: The possible role of RAMS11 in tumorigenesis, growth in the primary tumor and lymphatic, lung and liver metastases is not well characterised or discussed.

3. Mechanistic studies: The study is inadequate in characterizing the molecular mechanisms underlying various RAMS11-mediated phenotypes.

Major Concerns

1. Phenotypic characterisation: The role of RAMS11 in tumorigenesis, primary tumour growth and lymphatic, lung and liver metastases is not well characterised. Some additional analyses should be performed and the contributions of RAMS11 to these related but distinct processes (and clinical windows of treatment) should be examined and discussed.

Within 2 small datasets available to the authors, Amongst 148 differentially expressed DEGs between liver metastases versus primary/normal, 6 were associated with disease free survival (DFS) in TCGA and 1 of these 6 – RAMS11 was also prognostic for DFS in another small dataset.

In these 2 datasets, RAMS11 was identified as the top up-regulated novel lncRNA in colorectal liver metastases versus primary tumours.

Yet, in TCGA, RAMS11 was overexpressed in primary tumour versus normal tissue in colorectal cancer and also in 4 other cancers.

Experimentally, perturbation of RAMS11 affected phenotypes such as proliferation, invasion and anchorage-independent growth in vitro and lung metastases after tail vein injection in vivo.

This behooves the authors to examine the role of RAMS11 in tumorigenesis, primary tumour growth and its role in liver (and other organ) metastases:

a. Does overexpression of RAMS11 also contribute to poor prognosis or propensity for recurrence/metastases in other cancers, particularly these 4 cancers?

- b. Within the matched Kim dataset, a matched analyses should be done of RAMS11 expression in matched samples (amongst the cohort -- are the same tumors with relatively higher expression of RAMS11 in the primary, also likely to have relatively higher expression in the metastases?)
- c. With the discovery of the RAMS11 overexpression amongst liver tumors in the dataset and the liver being the most common and often first site of metastasis, the authors should investigate the effect of RAMS11 in experimental models of liver metastasis (i.e. intrasplenic or portal vein injection) rather than only tail vein injection. Alternatively, the effect of RAMS11 on orthotopic tumor growth (via cecal implantation/injection) and subsequent metastatic outgrowth could be investigated.
- d. Given the overexpression of RAMS11 in tumors (datasets) and effect on proliferation and growth in vitro, determination of whether RAMS11 affects (1) tumor development e.g. in capsult injection (2) tumor growth in xenografts should be evaluated.
- e. The authors should at least validate RAMS11 up-regulation by qPCR in their clinical samples to validate the results of their transcriptomic and meta-analysis. (including differential expression in matched metastases, tumor, normal tissue) and also evaluate it's expression in nodal disease.

2. Whilst the findings are interesting. The clinical value is rather overstated. There are many prognostic biomarkers in colorectal cancer and beyond MSI and BRAF mutation and to a lesser extent, some multigene panels (coloprint, oncotype Dx), few prognostic markers are used clinically because of robustness of results and constrained clinical utility of prognostic markers alone (without specific drug selection or stratification information beyond prognosis). In particular, Top2 inhibitors are not used routinely in colorectal cancer at all and except for an rather obscure clinical trial, it is not part of investigation in colorectal cancer as a drug of use. Notably, the TOP1 inhibitor irinotecan is used in almost all patients at some point with metastatic colorectal cancer. However, the effect of this RAMS11 on cell viability in irinotecan (fig 5b) is negligible. Perhaps, evaluating the effect of RAMS11 across a larger panel of cell lines on effect on 5FU/Oxaliplatin/irinotecan drugs used in colon cancer will be more relevant. Alternatively, examining RAMS11 in cancers where TOP2 inhibitors are used (e.g. breast or gastric or endometrial cancers) [even if it is not prognostic in those settings] but the mechanism is retained, will at least have some potential medium-term apparent clinical value/relevance and context which the authors are looking for.

3. Mechanistic insights: The authors sought to understand the mechanisms through which RAMS11 regulate TOP2 α protein expression. The authors performed experiments to demonstrate the interaction of RAMS11 and CBX4 with regards Top2 α promoter. The results explained the demonstrated resistance of RAMS11 expressing cells to TOP2 α inhibitors. However, what was interesting and regrettably not followed up in greater detail was what other potential targets could RAMS11 and CBX4 regulate in addition to TOP2 α that could further contribute to poorer prognosis. TOP2 α contributing to proliferation and invasion of colon cancer cells and other cancers have already been previously reported and not entirely novel and I feel that the authors' efforts could have been better directed towards identification of novel targets through the use of unbiased methods to identify novel RAMS11-CBX4 promoter targets instead of qPCR of a specific site.

Minor Concerns

1. Speculative statements should be removed e.g. page 10, "Overexpression of RAMS11 could be contributing to previous reports that these therapies alone or in combination continue to have low response rates and the treatment of metastatic disease remains essentially palliative" (no data to support this)
2. CCD18-Co is a fetal colonic line and might not be the most appropriate control. Just a correlation between the RAMS11 expression and metastatic capability (measured using associated phenotypes) of the panel of ATCC cell-lines could be potentially informative. For example, the SW620 cell-line was derived from lymph node metastases of the same patient as SW480. If there is no correlation, perhaps the extent of nuclear localization of RAMS11 differ between cell-lines.
3. In their over-expression studies, the extent of over-expression was extremely high, how did that compare to what was observed in the patient samples or TCGA data when normal tissues were compared to primary tumors or metastatic tissues? Would the observed phenotype resulting from over-expression be an artifact of exceedingly high over-expression?
4. Regarding in vivo experiments performed by the authors where they perturbed RAMS11 levels through CRISPR-KO, I am concerned that the experiments were not performed over-expression/gain-of-function experiments as the authors identified RAMS11 as an over-expressed lncRNA in aggressive CRC. Their current results were almost to be expected given the decreased in proliferation observed in RAMS11--KO cells. That being the case, a gain-of-function experiment should be performed. Loss-of-function studies demonstrate necessity (i.e RAMS11 is required for metastasis), gain-of-function studies demonstrate sufficiency (i.e. RAMS11 can promote metastasis).
5. Did the lungs metastases that grew out from WT cells showed increased RAMS11 expression?
6. Was there any impact on the expression of LINC01564 after deletion of RAMS11?
7. Were the observed phenotypes of invasion, migration and anchorage dependent growth dependent on CBX4 and TOP2 β ? If so the involvement of TOP2 β in such phenotypes have already been shown by others. The data only suggest somewhat that it is involved in resistance.
8. Instead of percentage viability, or relative viability, dose-response studies should be used to calculate difference in IC50 after over-expression of RAMS11 (fig. 5b), consistent with fig. 5c.

Reviewer #1 (Remarks to the Author):

In this study Silva-Fisher et al., identified lncRNAs associated with the development of metastatic colorectal cancer. They focused on one lncRNA that they named RAMS11. For the most part, the data is strong, the findings are novel and interesting, and the manuscript is well-written. The points below highlight some of the weaknesses and concerns that could be addressed. Overall, this is a nice study, the strengths outweigh the weaknesses and the impact of this study is considered high.

- We would like to thank the Reviewer for their positive comments and acknowledging the impact of our study.

Major Comments:

1. Fig. 1: The authors mention in the Fig. legend about 5'3' RACE but this data appears to be missing. Also, in Fig. 1C, are the numbers on the Y-scale (0, 2) no of reads, RPKM, FPKM, etc?

- We have clarified in the manuscript how we used RACE to determine the full-length *RAMS11* transcript and highlighted that the results can be found in Supplementary Table 2. We have also added a label to Fig. 1d labeled 5' and 3' RACE (blue) and RefSeq *NR_125841* (black) for the previously annotated lncRNA *LINC01564*. We have included Rebuttal Figure 1a UCSC screen shot of six clones that were amplified from the 5' and 3' end of *RAMS11* from the LoVo colon cancer cell line. In our previous Fig. 1C now Fig. 1d, the legend now indicates normalized coverage for Y-scale.

Rebuttal Figure 1 | *RAMS11* 5' and 3' RACE validation and size (a) *RAMS11* 5' and 3' RACE sequenced five exon transcript shown on UCSC browser. (b) RT-PCR amplified *RAMS11* in the LoVo cell line.

2. The authors need to mention in the text that *RAMS11* is *LINC01564*. Without this, the Ref Seq ID will be buried in Table S1, as it is now. In fact, it looks like *LINC01564* is a part of *RAMS11*? The 5' and 3' RACE data is not there. If it there, the authors should modify the text accordingly because in the current manuscript, it seems that *RAMS11* is a novel transcript. It would be important to determine the length of the *RAMS11* RNA by Northern blotting and/or RT-PCR for the full-length lncRNA.

- We apologize for the confusion. The existing annotated lncRNA, *LINC01564*, was reported as a three-exon gene. However, by leveraging the unbiased nature of transcriptome sequencing, followed by 5' and 3' RACE, we have determined that the lncRNA is actually a five-exon transcript, which we termed *RAMS11*. We have updated the manuscript (Figure 1d) to show that *RAMS11* is a five-exon gene and indicated how this compares to the existing three-exon annotation (*LINC01564*). In addition to the 5' and 3' RACE results (refer to previous comment and Rebuttal Figure 1a), we have RT-PCR amplified *RAMS11* in the LoVo cell line revealing a 986 bp transcript (Rebuttal Figure 1b), which corresponds with our human patient RNA-Seq data.

3. Does the expression of *RAMS11* correlate with patient survival in other cancer types in addition to COAD?

- We evaluated whether the expression of *RAMS11* correlated with patient survival in cancer types we previously found to have up-regulated *RAMS11* expression (see Supplementary Figure 3a). However, we did not find an association between *RAMS11* expression and patient survival in LUAD, LUSC, HNSC, and KIRP using TCGA datasets.

4. Fig. 1D: Strangely, the authors called the first exon on *RAMS11* the last exon.

- ***RAMS11* is encoded on the forward strand. Fig. 1d shows the full length transcript with the orientation indicated by the blue arrows.**

5. The order in which some of the data is presented is awkward. For example, would it not make more sense to show the efficacy of *RAMS11* CRISPR knockout and *RAMS11* overexpression data (Fig.s 3J and 3K) before the invasion data in Fig. 3 to demonstrate that these systems were effectively constructed before they provide data where these systems were used.

- **We appreciate this recommendation and have reorganized the data to show the knockdown and overexpression efficiency first.**

6. Along the same lines, they describe the results of Fig. 3 before the results of Fig. 2B-2G.

- **The results are now discussed in the same order as the Figures appear.**

7. The data on positive regulation of TOP2 α by *RAMS11* is very interesting and novel. It fits very well with the functional data in Fig. 5. Is there a positive correlation between *RAMS11* expression and TOP2 α in CRC patients and is there any link to patient survival? If not in CRC, could the authors check this in other cancer types?

- **We appreciate the reviewer finds the positive regulation of TOP2 α by *RAMS11* to be interesting and novel. We did not find a positive correlation between *RAMS11* and TOP2 α in the WUSTL, Kim, and TCGA cohorts or additional cancer types (LUAD, LUSC, HNSC, and KIRP using TCGA data). Additionally, we performed survival analysis and did not observe an association with patient survival for TOP2 α alone. Further, incorporating TOP2 α with *RAMS11* did not improve survival prediction in CRC (Rebuttal Figure 2).**

Rebuttal Figure 2 | TOP2 α is not associated with survival in TCGA dataset and combining *RAMS11* and TOP2 α does not improve the prognostic power compared with *RAMS11* alone. (a) Kaplan Meier curves for *RAMS11*, (b) TOP2 α , and (c) multivariate analysis combining *RAMS11* and TOP2 α (p-value and HR shown are between low *RAMS11*/low TOP2 α vs. high *RAMS11*/high TOP2 α groups).

Could they also check for correlation between *RAMS11*, TOP2 α and a CBX4-regulated transcriptome signature in CRC patients?

- **We appreciate the suggestion, however, the limited literature on CBX4 and TOP2 α regulated genes hindered our ability to generate a 'gold standard' *RAMS11*, TOP2 α , and CBX4 signature to evaluate across patients.**

8. Additionally, these analysis between *RAMS11*, TOP2 α , some CBX4-regulated genes and H3K4me3 could be done in the panel of CRC cell lines they already have. The data from these experiments could be useful in determining how broadly applicable the proposed model is.

- **We previously studied the association of *RAMS11*, TOP2 α , CBX4, and H3K4me3 in two metastatic colon cancer cell lines including LoVo and SW620 (Fig. 7). To assess how broadly applicable the proposed model is, we have expanded our analysis of CBX4 binding to *RAMS11* in two additional**

cell lines, SW480 and HCT116, which are two primary colon cancer cell lines. The negligible fold enrichment in these primary cell lines (Rebuttal Figure 3a and 3b) suggests the association of these genes may be specific to metastatic disease. This aligns with our patient data showing high *RAMS11* expression in metastatic tumors and its role in promoting an aggressive phenotype *in vitro* and *in vivo*. To further determine if this is specific to metastatic disease across cancers, we also assessed *RAMS11*-CBX4 binding in a lung metastatic cell line (HCC95). HCC95 cells have high levels of *RAMS11* and silencing *RAMS11* alters invasion (Supplementary Fig. 3b-d). Furthermore, we identified a 31-fold enrichment of CBX4 binding to *RAMS11* in HCC95 cells (Rebuttal Figure 3c). These results indicate that our model is potentially specific to metastatic disease and may also be applicable across other cancer types.

Rebuttal Figure 3: *RAMS11* RNA Immunoprecipitation (RIP) of CBX4. RIPs for *RAMS11* binding to CBX4 in (a) SW480, (b) HCT116, and (c) HCC95 cells.

9. Fig. 6C: To show the specificity of *RAMS11* interaction with CBX4, the authors can probe the immunoblot for a nonspecific protein such as Histone H3. One would expect to see the nonspecific protein in all 3 lanes.

- To show the specificity of *RAMS11* interacting with CBX4, we have additionally probed our immunoblot for a common negative control, SNRNP70. It is a well-known RNA binding protein that interacts with the snoRNA, *U1* (Rebuttal Figure 4), but should not bind to *RAMS11*. The input of the western blot shows high expression of SNRNP70 in our cells. However, our sense probe and our anti-sense (negative control) probe do not interact with SNRNP70 highlighting the specificity of *RAMS11* with CBX4. Further, CBX4 only binds to the *RAMS11* full-length RNA sense probe but not our negative control antisense probe.

Rebuttal Figure 4: RNA pull down of LoVo cells shows binding of *RAMS11* to CBX4 but not SNRNP70.

10. What are the regions of the *RAMS11* RNA that bind to CBX4? The authors can identify these regions by *in vitro* RNA pulldowns similar to Fig. 6C but using fragments of *RAMS11*.

- In order to identify the regions of *RAMS11* that bind to CBX4, we conducted *in vitro* RNA pulldown in LoVo cell line. We used four truncated *RAMS11* fragments (Supplementary Fig. 7a; Rebuttal Figure 5a). We re-validated our previous findings that full length (FL) *RAMS11* binds to CBX4 and revealed that nucleotides 600-959 of *RAMS11* are efficient to interact with CBX4 protein (Supplementary Fig. 7b; Rebuttal Figure 5b).

11. Experiments demonstrating the effect of RAMS11 expression on CBX4 binding to TOP2 α promoter may be strengthened by RAMS11 overexpression and rescue experiments in addition to the RAMS11 knockout.

- In addition to demonstrating the loss of CBX4 binding to the TOP2 α promoter in our CRISPR KO cell lines (Fig. 7d-g), we have added RAMS11 overexpression (OE) and rescue experiments into Figure 7. First, we performed RIP and ChIP of CBX4 using our HT29 RAMS11-overexpressing cell lines. Overexpressing RAMS11 in HT29 colon cancer cells increased binding of CBX4 to RAMS11 (Fig. 7h; Rebuttal Figure 6h) and increased binding of CBX4 (Fig. 7i; Rebuttal Figure 6i) and H3K4Me3 (Fig. 7j; Rebuttal Figure 6j) to the promoter of TOP2 α promoter. Second, we performed rescue experiments by re-expressing RAMS11 in our CRISPR KO cell lines. We found that reintroducing RAMS11 into the CRISPR KO cell lines rescued the binding of CBX4 and H3K4me3 to the TOP2 α promoter (Fig. 7k and l, Rebuttal Figure 6k and l).

Rebuttal Figure 5 (Manuscript Supplementary Fig. 7): RNA pull down of truncated RAMS11 fragments shows nucleotides 600-959 binding to CBX4. (a) RAMS11 five-exon transcript (top) and four created truncated RAMS11 fragments. (b) Western blot of CBX4 of RNA pull down with input, full length (FL) and four truncated RAMS11 fragments showing interaction at 600-959 and no binding

Rebuttal Figure 6 (Manuscript Fig. 7) | *RAMS11* binds to Chromobox 4 (CBX4) to regulate expression of TOP2 α mRNA and protein (a) RNA immunoprecipitation (RIP) shows binding of *RAMS11* to CBX4 and not negative control IgG in LoVo and (b) SW620 cells. (c) RNA pulldown of 5-Bromo-UTP full length *RAMS11* probe showing binding of CBX4 by western blot in LoVo and SW620 cells. (d-g) Decreased binding of CBX4 and active histone mark H3K4me3 at TOP2 α promoter with silenced *RAMS11* expression Chromatin immunoprecipitation (ChIP) assay. (h) RIP showing increased binding of *RAMS11* to CBX4 in HT29 *RAMS11* overexpressing cells. (i and j) ChIP of CBX4 and H3K4me3 shows increased binding to TOP2 α promoter in HT29 *RAMS11* overexpressing cells. (k and l) ChIP of CBX4 and H3K4me3 in CRISPR KO cells with *RAMS11* overexpression (OE) rescue TOP2 α promoter enrichment. (m) Protein expression of TOP2 α and CBX4 in LoVo (top) and SW620 cell lines (bottom). Band intensities were quantified from the digital image in ImageJ and are shown normalized to the Wild Type lane for each target. Fold change normalized expression to Actin is shown below gel. All data analyzed by t-test. * p < 0.05 **p > 0.005, # p < 0.0005.

12. The authors should address the apparent decrease in CBX4 expression after *RAMS11* knockout in Figures 6H and 6I.

- **We have repeated the western blots for new Fig. 7m showing LoVo CRISPR lines protein expression for TOP2 α and CBX4. Band intensities were quantified from the digital image in ImageJ and are shown normalized to the wild type lane for each target. Fold change of normalized expression to Actin is shown below gel. We did not observe a significant decrease in CBX4 protein expression in either the CRISPR KO cells or SW620 cells after transient silencing of *RAMS11*. We also assessed protein stability by CHX treatment and mRNA stability by Actinomycin treatment. We did not observe any significant changes in CBX4 protein or mRNA stability between the knockdown and cells.**

Minor Comments:

1. The decrease in TOP2 α expression in LoVo western blot is not very striking (Fig. 6H).

We have repeated the western blots (now Fig. 7m in the revised manuscript) showing TOP2 α protein levels in LoVo CRISPR KO cells. We quantified the bands to show a significant decrease in TOP2 α seen by relative change normalized to Actin.

2. The title could be more informative – they mainly study only one lncRNA, so it may be better to include that lncRNA in the title.

- **We appreciate this suggestion and have updated the title as follows: “Transcriptome analysis reveals *RAMS11*, a long non-coding RNA promoting metastatic colorectal cancer.”**

3. Could include a reference for “response to second line treatment is even less effective than first line” (Introduction, end of first paragraph).

- **We have updated the text to include the reference (PMID: 5083064).**

4. Second paragraph introduction: “significance of lncRNAs can be exemplified by their use as diagnostic, prognostic, and predictive biomarkers, and potential use as therapeutic targets.” Change to: “diagnostic, prognostic, predictive biomarkers, and potential use as therapeutic targets”

- **We appreciate this suggestion and updated the text accordingly.**

5. Third paragraph introduction: include reference for “lncRNAs have been shown to promote tumor progression”

- **We have updated the text to include three references pertaining to lncRNAs associated with tumor progression (PMID: 0900417, PMID: 0962766, PMID: 26617879).**

6. Page numbers and/or line numbers are missing

- **The line numbers have been added to the manuscript.**

7. Fig. 6C: Change Ant- to Anti-

- **This has been updated in the manuscript.**

Reviewer #2 (Remarks to the Author):

In the manuscript by Silva-Fisher et al., the authors performed transcriptomic and meta-analysis of lncRNAs expression in two (small) independent cohorts of CRC patients and identified RAMS11 as a novel lncRNA that promotes metastasis/tumorigenesis in CRC and potentially other cancers. They further performed in vitro and in vivo studies to further characterize phenotypes mediated by RAMS11 that could contribute to tumor progression and metastasis and identified RAMS11 as a novel interacting partner of CBX4 which in turns mediate binding of RAMS11-CBX4 complex to the promoter of TOP2 α . The study is of novelty given the authors' finding that RAMS11 is prognostic and indicative of resistance to TOP2 α inhibitors.

- **We would like to thank the Reviewer for complementing the novelty of our manuscript.**

Major Concerns

1. Phenotypic characterization: The role of RAMS11 in tumorigenesis, primary tumour growth and lymphatic, lung and liver metastases is not well characterized. Some additional analyses should be performed and the contributions of RAMS11 to these related but distinct processes (and clinical windows of treatment???) should be examined and discussed. Within 2 small datasets available to the authors, Amongst 148 differentially expressed DEGs between liver metastases versus primary/normal, 6 were associated with disease free survival (DFS) in TCGA and 1 of these 6 – RAMS11 was also prognostic for DFS in another small dataset. In these 2 datasets, RAMS11 was identified as the top up-regulated novel lncRNA in colorectal liver metastases versus primary tumors. Yet, in TCGA, RAMS11 was overexpressed in primary tumour versus normal tissue in colorectal cancer and also in 4 other cancers. Experimentally, perturbation of RAMS11 affected phenotypes such as proliferation, invasion and anchorage-independent growth in vitro and lung metastases after tail vein injection in vivo. This behooves the authors to examine the role of RAMS11 in tumorigenesis, primary tumour growth and its role in liver (and other organ) metastases:

a. Does overexpression of RAMS11 also contribute to poor prognosis or propensity for recurrence/metastases in other cancers, particularly these 4 cancers?

- **Survival analysis in LUAD, LUSC, HNSC, and KIRP using TCGA datasets and did not show an association between *RAMS11* expression and patient outcome. However, overexpression of *RAMS11* appears to have a propensity for metastasis in other cancers. For instance, Supplementary Fig. 3a shows that *RAMS11* is overexpressed in primary tumors versus normal tissue in LUSC, LUAD, HNSC, and KIRP. We did additional phenotype characterization on the invasive capabilities of *RAMS11* in lung cancer using a LUSC (HCC95) and LUAD (A549) cell line. We show that silencing *RAMS11* decreases invasion in both HCC95 cells (siRNA1 and siRNA2 $p < 0.05$, Supplementary Fig. 3b-d) and A549 cells (siRNA1 and siRNA2 $p < 0.005$, Supplementary Fig. 3e-f).**

b. Within the matched Kim dataset, a matched analyses should be done of RAMS11 expression in matched samples (amongst the cohort -- are the same tumors with relatively higher expression of RAMS11 in the primary, also likely to have relatively higher expression in the metastases?

- **We apologize that we were not clear in our methods, but the differentially expressed genes were inferred using a matched analysis in Kim dataset. We have updated the manuscript to indicate that these were matched. We have also manually investigated the expression of *RAMS11* in the Kim dataset and found that, in line with our differential expression results, *RAMS11* has higher expression in the metastatic specimen when it is also highly expressed in the matched primary (Rebuttal Figure 7).**

Rebuttal Figure 7 | *RAMS11* expression in matched Kim dataset (a) Expression and (b) fold change between metastasis and primary, metastasis and normal, and primary and normal for calculated for individual patients. Normal (N), Primary (P), and Metastasis (M). Dashed horizontal line indicates fold change = 1 (i.e. no change).

c. With the discovery of the *RAMS11* overexpression amongst liver tumors in the dataset and the liver being the most common and often first site of metastasis, the authors should investigate the effect of *RAMS11* in experimental models of liver metastasis (i.e. intrasplenic or portal vein injection) rather than only tail vein injection. Alternatively, the effect of *RAMS11* on orthotopic tumor growth (via cecal implantation/injection) and subsequent metastatic outgrowth could be investigated.

- We appreciate the suggestion to investigate the effect of *RAMS11* in an experimental model of liver metastasis. To address this, we have added data (Fig. 5; Rebuttal Figure 8) using a hemisplenectomy (intrasplenic) model. This is a relevant tumor model for assessing the effect of *RAMS11* on liver metastasis *in vivo*. Overall, we found that *RAMS11* CRISPR KO cells show a significant decrease in liver metastasis compared to the wild type cell (Fig. 5a and b and Rebuttal Figure 8a and b). We excised all mouse livers and validated a decrease of (i) liver metastasis in *RAMS11* CRISPR KO cell-injected tumors (Fig. 5c; Rebuttal Figure 8c), (ii) liver weights (Fig. 5d; Rebuttal Figure 8d), and (iii) overall liver metastasis area (Fig. 5e; Rebuttal Figure 8e). Decreased tumor burden and proliferation in *RAMS11* CRISPR KO cell-injected livers was further determined by hematoxylin and eosin (H&E) and Ki67 staining (Fig. 5f; Rebuttal Figure 8f).
- Overall, we have shown that *RAMS11* effects primary tumor growth and metastasis via three mouse models: (1) a subcutaneous model to study tumor growth and metastases (Fig. 3), (2) a tail vein injection model to study the development of lung metastases (Fig. 4), and (3) a hemi-splenectomy model to study the development of liver metastases (Fig. 5).

Rebuttal Figure 8 (Manuscript Fig. 5) | *RAMS11* induces metastasis via hemisplenectomy mouse model. (a) Representative mice showing no liver metastasis in *RAMS11* CRISPR KO cell-injected mice by BLI. (b) *RAMS11* CRISPR KO cells show a significant decrease in liver metastasis by Day 21. (c) Day 21 ex vivo mouse livers show decreased metastasis in *RAMS11* CRISPR KO mice by BLI. Wild type mice had (d) increased liver weights and (e) liver metastasis compared to CRISPR KO cell lines. (f) Hemotoxylin and Eosin stain of livers showing metastasis (M) and levels of Ki67 stain. Data shown as mean ± SEM. *p < 0.05, ** p < 0.005, # p < 0.0005.

d. Given the overexpression of *RAMS11* in tumors (datasets) and effect on proliferation and growth in vitro, determination of whether *RAMS11* affects (1) tumor development e.g. in capsult injection (2) tumor growth in xenografts should be evaluated.

- **Our apologies for not clearly highlighting our data evaluating whether overexpression of *RAMS11* promotes tumor growth *in vivo* in our initial submission. In our revised manuscript, Fig. 3 shows that our CRISPR KO cells decrease tumor volume when compared to wild type cells using a subcutaneous model.**

e. The authors should at least validate *RAMS11* up-regulation by qPCR in their clinical samples to validate the results of their transcriptomic and meta-analysis. (Including differential expression in matched metastases, tumor, normal tissue) and also evaluate it's expression in nodal disease.

- **We isolated RNA from matched 12 normal, 14 primary, and 14 liver metastatic tissues from colon cancer patients. We validated *RAMS11* up-regulation in the liver metastasis tissues (p value=0.0059) compared to normal tissues and also up-regulation of liver metastasis tissues**

compared to primary tissues (p value=0.015, Supplementary Fig. 1b and Rebuttal Figure 9). These results further validate our findings in the first submission.

Rebuttal Figure 9 (Manuscript Supplementary Fig. 1b) | Validation of *RAMS11* up-regulation in clinical samples by RT-qPCR. RT-qPCR validation of 14 additional matched (normal, primary, metastatic) patient samples showing increased *RAMS11* expression in metastatic samples. *p value > 0.05, ** p value > 0.005.

2. Whilst the findings are interesting. The clinical value is rather overstated. There are many prognostic biomarkers in colorectal cancer and beyond MSI and BRAF mutation and to a lesser extent, some multigene panels (coloprint, oncotype Dx), few prognostic markers are used clinically because of robustness of results and constrained clinical utility of prognostic markers alone (without specific drug selection or stratification information beyond prognosis). In particular, Top2 inhibitors are not used routinely in colorectal cancer at all and except for an rather obscure clinical trial, it is not part of investigation in colorectal cancer as a drug of use. Notably, the TOP1 inhibitor irinotecan is used in almost all patients at some point with metastatic colorectal cancer. However, the effect of this *RAMS11* on cell viability in irinotecan (Fig 5b) is negligible. Perhaps, evaluating the effect of *RAMS11* across a larger panel of cell lines on effect on *5FU/Oxaliplatin/irinotecan* drugs used in colon cancer will be more relevant.

- **Our revised manuscript has new data across additional cell lines assessing the effects of *RAMS11* on drug sensitivity. We previously show (Fig. 6b) that treating HT29 cells overexpressing *RAMS11* with Irinotecan or Oxaliplatin (Supplementary Table 3) did not significantly affect cellular viability in comparison to the empty vector control cells. However, we did see a significant change in cellular viability with Floxiruidine (5-FU) treatment (Supplementary Fig. 4c). We expanded our evaluation of these drugs in two additional cell lines, *RAMS11* LoVo CRISPR KO cells and SW620 cells with transient knockdown of *RAMS11*. The *RAMS11* CRISPR KO cells had a 1.7-fold and 5.8-fold increase in 5-FU sensitivity, in CRISPR1 and CRISPR2, respectively, compared to wild type cells (Supplementary Fig. 5a; Rebuttal Figure 10a). Similarly, SW620 cells with transiently silenced *RAMS11* had a greater than 1.5-fold increase in 5-FU sensitivity (siRNA1 Fold > 1.53, siRNA2 Fold > 1.59) relative to scrambled control (Supplementary Fig. 5b; Rebuttal Figure 10b). We did not see a significant effect of cellular viability for Irinotecan or Oxaliplatin treatment using HT29 *RAMS11* overexpressing or LoVo *RAMS11* CRISPR KO cells. (Supplementary Fig 5c-e; Rebuttal Figure 10c-e). SW620 cells with silenced *RAMS11* also did not have a significant effect of cellular viability for Oxaliplatin treatment but we do see a significant sensitivity to Irinotecan (siRNA1 Fold > 3.17, siRNA2 Fold > 11.8, Supplementary Fig. 5f; Rebuttal Figure 10f). Overall, we determined that *RAMS11* expression does have an effect on 5-FU in three cell lines (HT29, LoVo, and SW620) and on Irinotecan in only SW620 cells.**

Rebuttal Figure 10 (Supplementary Fig. 5) | IC50s of clinically used drugs for colorectal cancer treatment. *RAMS11* CRISPR KO cell lines and SW620 cells with silenced *RAMS11* treated with 5-FU (a and b), Oxaliplatin (c and d), and Irinotecan (e and f) drug treatments. *Fold > 1.5, ** Fold > 5.

Alternatively, examining *RAMS11* in cancers where TOP2 inhibitors are used (e.g. breast or gastric or endometrial cancers) [even if it is not prognostic in those settings] but the mechanism is retained, will at least have some potential medium-term apparent clinical value/relevance and context which the authors are looking for.

- **We appreciate this suggestion and while the examination of *RAMS11* in other cancers where TOP2 α inhibitors are used would be highly informative, our pan-cancer analysis does not show high *RAMS11* expression in these other cancer types. Further, for this study we specifically chose to focus on metastatic colon cancer due to the limited knowledge lncRNA regulation in mCRC coupled with our access to a unique CRC patient cohort.**

3. Mechanistic insights: The authors sought to understand the mechanisms through which *RAMS11* regulate TOP2 α protein expression. The authors performed experiments to demonstrate the interaction of *RAMS11* and CBX4 with regards TOP2 α promoter. The results explained the demonstrated resistance of *RAMS11* expressing cells to TOP2 α inhibitors. However, what was interesting and regrettably not followed up in greater detail was what other potential targets could *RAMS11* and CBX4 regulate in addition to TOP2 α that could further contribute to poorer prognosis. TOP2 α contributing to proliferation and invasion of colon cancer cells and other cancers have already been previously reported and not entirely novel and I feel that the authors' efforts could have been better directed towards identification of novel targets through the use of unbiased methods to identify *novel RAMS11-CBX4 promoter targets* instead of qPCR of a specific site.

- **We appreciate this suggestion. Given the novelty of our finding that a previously uncharacterized lncRNA interacts with CBX4 to promote mCRC, we wanted to focus on a specific target gene to comprehensively dissect lncRNA dependent regulation. We felt that pursuing TOP2 α as a target gene was compelling since it since our unbiased NIH drug panel revealed that *RAMS11* expression affected cellular sensitivity in topoisomerase inhibitors. However, we feel expanding this analysis to identify additional downstream targets would be a logical and compelling follow-up study.**

Minor Concerns

1. Speculative statements should be removed e.g. page 10, "Overexpression of RAMS11 could be contributing to previous reports that these therapies alone or in combination continue to have low response rates and the treatment of metastatic disease remains essentially palliative" (no data to support this)

- **We have removed this statement.**

2. CCD18-Co is a fetal colonic line and might not be the most appropriate control. Just a correlation between the RAMS11 expression and metastatic capability (measured using associated phenotypes) of the panel of ATCC cell-lines could be potentially informative. For example, the SW620 cell-line was derived from lymph node metastases of the same patient as SW480. If there is no correlation, perhaps the extent of nuclear localization of RAMS11 differ between cell-lines.

- **We acknowledge that cell lines are not always ideal models despite their utility. As suggested by the reviewer, we compared the SW480 and SW620 pair of cell lines. We found that there is less binding of CBX4 to RAMS11 in the primary SW480 (refer to Reviewer 1 Comment 8 Rebuttal Figure 3a) as compared to the metastatic cell line SW620. This supports our current hypothesis that higher expression of RAMS11 interacts with CBX4 to regulate genes promoting metastatic disease. In addition to using the CCD18-Co control line for expression, we also generated 3 models for assessing aggressive phenotypes that include RAMS11 CRISPR KO lines (LoVo) (Fig. 2a), RAMS11 OE lines (HT29) (Fig. 2b), and transient silencing with siRNAs (SW620). Further, our access to patient data of matched normal, primary and metastatic tissues provides compelling evidence and clinical relevance of the association of RAMS11 with metastasis - more so than a cell line panel.**

3. In their over-expression studies, the extent of over-expression was extremely high, how did that compare to what was observed in the patient samples or TCGA data when normal tissues were compared to primary tumors or metastatic tissues? Would the observed phenotype resulting from over-expression be an artifact of exceedingly high over-expression?

- **We assessed expression by qPCR of RAMS11 in a validation cohort of patients samples from our transcriptome sequencing data and see on average 10-fold overexpression of RAMS11 (Supplementary Fig. 1b). We agree that our cells have extremely high RAMS11 overexpression. However, to rule out the possibility that the results are the artifact of high over-expression, we also perform knockdown experiments. We found that two independent methods for silencing RAMS11 (CRISPR KO and transient silencing) generated data that corroborated our overexpression phenotypes (albeit they have opposite effects).**

4. Regarding in vivo experiments performed by the authors where they perturbed RAMS11 levels through CRISPR-KO, I am concerned that the experiments were not performed over-expression/gain-of-function experiments as the authors identified RAMS11 as an over-expressed lncRNA in aggressive CRC. Their current results were almost to be expected given the decreased in proliferation observed in RAMS11^{-/-}-KO cells. That being the case, a gain-of-function experiment should be performed. Loss-of-function studies demonstrate necessity (i.e RAMS11 is required for metastasis), gain-of-function studies demonstrate sufficiency (i.e. RAMS11 can promote metastasis).

- **Although, we agree with the Reviewer about the importance of doing gain-of-function studies to show that RAMS11 promotes metastasis, we have evidence of its effect on tumor growth and metastasis using three mouse models (Figs. 3-5) using the CRISPR lines.**

5. Did the lungs metastases that grew out from WT cells showed increased RAMS11 expression?

- **We isolated RNA from the lungs from 2 mice injected with the wild type cell lines injected by tail vein and did see expression of RAMS11 (Rebuttal Figure 11).**

Rebuttal Figure 11 | *RAMS11* expression from lungs of mice injected with LoVo wild type cells via tail vein model.

6. Was there any impact on the expression of *LINC01564* after deletion of *RAMS11*?

- **We apologize for the confusion. *LINC01564* is apart of *RAMS11*. Please see the full explanation above in Reviewer 1 Comments 1 and 2.**

7. Were the observed phenotypes of invasion, migration and anchorage dependent growth dependent on *CBX4* and *TOP2α*? If so the involvement of *TOP2α* in such phenotypes have already been shown by others. The data only suggest somewhat that it is involved in resistance.

- **The observed phenotypes are due to *RAMS11*-dependent *CBX4* transcriptional regulation of *TOP2α*. We appreciate that others may have demonstrated *TOP2α* is involved in these phenotypes. However, our study uniquely shows that this is dependent on a previously uncharacterized lncRNA interacting with *CBX4* to activate *TOP2α*.**

8. Instead of percentage viability, or relative viability, dose-response studies should be used to calculate difference in *IC50* after over-expression of *RAMS11* (Fig. 5b), consistent with Fig. 5c.

- **We utilized the NIH Drug panel of around 200 drugs to identify any possible targets of *RAMS11*, which included only three drug concentrations, which is not sufficient for creating *IC50*s. We used this panel as a screening to determine what, if any, drugs showed sensitivity with changes in expression of *RAMS11*. The drugs that showed statistical differences were then used to determine the *IC50*.**

REVIEWERS' COMMENTS:

Reviewer #1 (Remarks to the Author):

The authors have addressed all of my critiques. It was a pleasure reading this very nice study.

Reviewer #3 (Replacement for Reviewer#2, Remarks to the Author):

Overall, the authors provided a revised version in full of the previously submitted manuscript. This version reads more clearly and has significantly and substantially improved. In particular, this reviewer feels quite compelled by the experiments and data clarifying the liver metastasis colonization effects in vivo (Intra-splenic injections). Similarly, i) A clarification that RAMS11 is relevant in certain treatments but not the previously reported is adequate. ii) the expression data in primary tumors and metastasis is now clearer. Now the authors provide more interesting observations positioning their molecular-based results in the right clinical context.

In summary, the manuscript represents a relevant piece of information and has largely improved from the review process. Although not all points have been addressed, compelling data sustains the authors' claims. At this stage, this reviewer only has two small sticky points.

1. Minor point 5. The authors agree with the Reviewer about the importance of doing gain-of-function studies to show that RAMS11 promotes metastasis, they have evidence of its effect on tumor growth and metastasis using three mouse models (Figs. 3-5) using the CRISPR lines. Indeed, this shows that RAMS11 is necessary for these processes. Yet, they still do not provide data on sufficiency. This is a must to sustain some of the claims. This limitation must be acknowledged and the discussion mention it. Otherwise, this overstates the findings.

2. Another minor point requested whether the lungs metastases that grew out from WT cells showed increased RAMS11 expression or not. The authors provide isolated RNA from the lungs from 2 mice injected with the wild type cell lines injected by tail vein and did see expression of RAMS11 (Rebuttal Figure 11). Yet the authors did not confirm increased RAMS11 expression, at least in a robust manner. This is odd and puzzling. If the magnitude is small (as inferred from the dCTs) and not significant, the statements should be tuned down.

Response to Referees

Reviewer #3:

Minor point 5. The authors agree with the Reviewer about the importance of doing gain-of-function studies to show that RAMS11 promotes metastasis, they have evidence of its effect on tumor growth and metastasis using three mouse models (Figs. 3-5) using the CRISPR lines. Indeed, this shows that RAMS11 is necessary for these processes. Yet, they still do not provide data on sufficiency. This a must to sustain some of the claims. This limitation must be acknowledged and the discussion mention it. Otherwise, this overstates the findings.

We have acknowledged this in the manuscript in the discussion section.

Another minor point requested whether the lungs metastases that grew out from WT cells showed increased RAMS11 expression or not. The authors provide isolated RNA from the lungs from 2 mice injected with the wild type cell lines injected by tail vein and did see expression of RAMS11 (Rebuttal Figure 11). Yet the authors did not confirm increased RAMS11 expression, at least in a robust manner. This is odd and puzzling. If the magnitude is small (as inferred from the dCTs) and not significant, the statements should be tuned down.

This has been toned down.